# SCALABLE DISCRETE DIFFUSION SAMPLERS: COMBINATORIAL OPTIMIZATION AND STATISTICAL PHYSICS

**Sebastian Sanokowski**[1] [*]    **Wilhelm Berghammer** [1]    **Martin Ennemoser** [1]

**Haoyu Peter Wang** [2]    **Sepp Hochreiter** [1,3]    **Sebastian Lehner** [1]

[1] ELLIS Unit Linz, LIT AI Lab, Johannes Kepler University Linz, Austria
[2] Department of Electrical & Computer Engineering, Georgia Institute of Technology
[3] NXAI Lab & NXAI GmbH, Linz, Austria

## ABSTRACT

Learning to sample from complex unnormalized distributions over discrete domains emerged as a promising research direction with applications in statistical physics, variational inference, and combinatorial optimization. Recent work has demonstrated the potential of diffusion models in this domain. However, existing methods face limitations in memory scaling and thus the number of attainable diffusion steps since they require backpropagation through the entire generative process. To overcome these limitations we introduce two novel training methods for discrete diffusion samplers, one grounded in the policy gradient theorem and the other one leveraging Self-Normalized Neural Importance Sampling (SN-NIS). These methods yield memory-efficient training and achieve state-of-the-art results in unsupervised combinatorial optimization. Numerous scientific applications additionally require the ability of unbiased sampling. We introduce adaptations of SN-NIS and Neural Markov Chain Monte Carlo that enable for the first time the application of discrete diffusion models to this problem. We validate our methods on Ising model benchmarks and find that they outperform popular autoregressive approaches. Our work opens new avenues for applying diffusion models to a wide range of scientific applications in discrete domains that were hitherto restricted to exact likelihood models.

## 1 INTRODUCTION

Sampling from unnormalized distributions is crucial in a wide range of scientific domains, including statistical physics, variational inference, and combinatorial optimization (CO) (Wu et al., 2019; Shih & Ermon, 2020; Hibat-Allah et al., 2021). We refer to research on using neural networks to learn how to sample unnormalized distributions as Neural Probabilistic Optimization (NPO). In NPO, a target distribution is approximated using a probability distribution that is parameterized by a neural network. Hence, the goal is to learn an approximate distribution in a setting, where only unnormalized sample probabilities can be calculated. Importantly, no samples from the target distribution are available, i.e. we are working in the data-free problem setting. In the following, we consider binary state variables $X \in \{0,1\}^N$, where $N$ represents the system size. The unnormalized target distribution is typically implicitly defined by an accessible energy function $H : \{0,1\}^N \to \mathbb{R}$. The target distribution is defined to be the corresponding Boltzmann distribution:

$$p_B(X) = \frac{\exp\left(-\beta H(X)\right)}{\mathcal{Z}}, \qquad \text{where} \quad \mathcal{Z} = \sum_X \exp\left(-\beta H(X)\right). \tag{1}$$

---

[*]Code available at: `https://github.com/ml-jku/DIffUCO`.
Correspondance to `sanokowski[at]ml.jku.at`

Here $\beta := 1/\mathcal{T}$ is the inverse temperature, and $\mathcal{Z}$ is the partition sum that normalizes the distribution. An analogous formulation applies to continuous problem domains. Unbiased sampling from this distribution is typically computationally expensive due to the exponential number ($2^N$) of states. Sampling techniques, such as Markov Chain Monte Carlo (Metropolis et al., 1953) are employed with great success in applications in statistical physics. Nevertheless, their applicability is typically limited due to issues related to Markov chains getting stuck in local minima and large autocorrelation times (Nicoli et al., 2020; McNaughton et al., 2020). Recently, the application of deep generative models has gained increasing attention as an approach to this problem. Initial methods in NPO relied on exact likelihood models where $q_\theta(X)$ could be efficiently evaluated. Boltzmann Generators (Noé & Wu, 2018) are a notable example in the continuous setting, using normalizing flows to approximate Boltzmann distributions for molecular configurations. In the discrete setting, Wu et al. (2019); Hibat-Allah et al. (2021) use autoregressive models to approximate Boltzmann distributions of spin systems in the context of statistical and condensed matter physics. Inspired by the success of diffusion models (Sohl-Dickstein et al., 2015; Ho et al., 2020) in image generation, there is growing interest in so-called diffusion samplers where these models are applied to NPO problems in discrete (Sanokowski et al., 2024) and continuous settings (Zhang & Chen, 2022). Diffusion models are particularly intriguing in the discrete setting due to the lack of viable alternatives. Normalizing flows, which are a popular choice for continuous problems, cannot be directly applied in discrete settings, leaving autoregressive models as the most popular alternative. However, autoregressive approaches face significant limitations. They become computationally prohibitive as the system size grows. There are complexity theoretical results (Lin et al., 2021) and empirical results (Sanokowski et al., 2024) that suggest that they are less efficient distribution learners than latent variable models like diffusion models. Consequently, we consider diffusion models as a more promising approach to discrete NPO. However, existing diffusion-based methods for sampling on discrete domains face two major challenges:

1. *Memory Scaling:* They rely on a loss that is based on the reverse Kullback–Leibler (KL) divergence which necessitates that the entire diffusion trajectory is kept in memory for backpropagation (see Sec. 2.1). This linear memory scaling limits the number of applicable diffusion steps and hence the achievable model performance. This is in sharp contrast to diffusion models in e.g. image generation, which benefit from the capability of using a large number of diffusion steps.

2. *Unbiased Sampling:* For many scientific applications, a learned distribution is only valuable if it allows for unbiased sampling, i.e., the unbiased computation of expectation values. Autoregressive models allow this through importance sampling or Markov Chain Monte Carlo methods based on their exact sample likelihoods. However, unbiased sampling with approximate likelihood models on discrete domains remains so far unexplored.

We introduce our method *Scalable Discrete Diffusion Sampler* (SDDS) by proposing two novel training methods in Sec. 3.1 to address the memory scaling issue and the resulting limitation on the number of diffusion steps in NPO applications of discrete diffusion models:

1. *reverse KL objective*: Employs the policy gradient theorem for minimization of the reverse KL divergence and integrates Reinforcement Learning (RL) techniques to mitigate the aforementioned linear memory scaling.

2. *forward KL objective*: Adapts Self-Normalized Neural Importance Sampling to obtain asymptotically unbiased gradient estimates for the forward KL divergence. This approach mitigates the linear memory scaling by using Monte Carlo estimates of the objective across diffusion steps.

In Sec. 5.1, we compare our proposed objectives to previous approaches and demonstrate that the reverse KL-based objective achieves new state-of-the-art results on 6 out of 7 unsupervised Combinatorial Optimization (UCO) benchmarks and is on par on one benchmark.
Secondly, to eliminate bias in the learned distribution, we extend two established methods - Self-Normalized Neural Importance Sampling (SN-NIS) and Neural Markov Chain Monte Carlo (NM-CMC) - to be applicable to approximate likelihood models such as diffusion models. We introduce these methods in Sec.3.2 and validate their effectiveness using the Ising model in Sec.5.2, highlighting the advantages of diffusion models over autoregressive models. Our experiments show that the forward KL divergence-based objective excels in unbiased sampling. We hypothesize that this is due to its mass-covering property. Our experiments show that the mass-covering property is also

beneficial in UCO when sampling many solutions from the model to obtain an optimal solution. Conversely, the reverse KL-based objective performs better in UCO contexts where only a few solutions are sampled or when a good average solution quality is prioritized.

## 2  PRELIMINARY: NEURAL PROBABILISTIC OPTIMIZATION

The goal of NPO is to approximate a known target probability distribution $p_B(X)$ using a probabilistic model parameterized by a neural network. This technique leverages the flexibility and expressive power of neural networks to model complex distributions. The objective is to train the neural network to represent a probability distribution $q_\theta(X)$ that approximates the target distribution without requiring explicit data from the target distribution. This approximation can generally be achieved by minimizing a divergence between the two distributions. One class of divergences used for this purpose are alpha divergences (Minka et al., 2005; Amari, 2012):

$$D_\alpha(p_B(X)\|q_\theta(X)) = -\frac{\int p_B(X)^\alpha q_\theta(X)^{1-\alpha}dX}{\alpha\,(1-\alpha)}$$

By selecting a specific value of $\alpha$, this divergence can be used as a loss function for training the model, and the choice of $\alpha$ influences the bias of the learned distribution. For instance, for $\alpha \leq 0$ the resulting distribution is mode seeking, which means the model focuses on the most probable modes of the target distribution, potentially ignoring less probable regions. Whereas, for $\alpha \geq 1$ it is mass-covering, meaning the model spreads its probability mass to cover more of the state space, including less probable regions. As $\alpha \to 1$ the divergence equals the forward Kullback-Leibler divergence (fKL) $D_{KL}(p_B(X)\,\|\,q_\theta(X))$ and as $\alpha \to 0$ it equals the reverse Kullback-Leibler divergence (rKL) $D_{KL}(q_\theta(X)\,\|\,p_B(X))$ (Minka et al., 2005). The two divergences, rKL and fKL are particularly convenient in this context due to the *product rule of logarithms* that we utilize in this paper to realize diffusion models with more diffusion steps (see Sec. 3.1).

### 2.1  DISCRETE DIFFUSION MODELS FOR NEURAL PROBABILISTIC OPTIMIZATION

In discrete time diffusion models, a *forward diffusion process* transforms the target distribution $p_B(X_0)$ into a *stationary distribution* $q(X_T)$ through iterative sampling of a *noise distribution* $p(X_t|X_{t-1})$ where $t \in \{1, T\}$ for a total of $T$ iterations. The diffusion model is supposed to model the reverse process, i.e. to map samples $X_T \sim q(X_T)$ to $X_0 \sim p_B(X_0)$ by iteratively sampling $q_\theta(X_{t-1}|X_t)$. The probability of a diffusion path $X_{0:T} = (X_0, ..., X_T)$ of the reverse process can be calculated with $q_\theta(X_{0:T}) = q(X_T)\prod_{t=1}^{T} q_\theta(X_{t-1}|X_t)$ and $q_\theta(X_{t-1}|X_t)$ is chosen so that samples $X_{0:T} \sim q_\theta(X_{0:T})$ can be efficiently drawn. Usually, in the reverse process, the diffusion model is explicitly conditioned on the diffusion step $t$, such that the distribution of the reverse diffusion step can be written as $q_\theta(X_{t-1}|X_t, t)$. However, in the following, we will drop the dependence on $t$ to simplify the notation. The unnormalized probability of a diffusion path of the forward process can be calculated with $\widehat{p}(X_{0:T}) = \widehat{p_B}(X_0)\prod_{t=1}^{T} p(X_t|X_{t-1})$. In the data-free setting samples $X_{0:T} \sim p(X_{0:T})$ are not available. Sanokowski et al. (2024) invoke the Data Processing Inequality to introduce diffusion models in discrete NPO by proposing to use the rKL of joint probabilities $D_{KL}(q_\theta(X_{0:T})\,\|\,p(X_{0:T}))$ as a tractable upper bound of the rKL of the marginals $D_{KL}(q_\theta(X_0)\,\|\,p_B(X_0))$. They further simplify this objective to express it in the following form:

$$\begin{aligned}
\mathcal{T}\,D_{KL}\left(q_\theta(X_{0:T})\,\|\,p(X_{0:T})\right) = &-\mathcal{T}\cdot\sum_{t=1}^{T}\mathbb{E}_{X_{t:T}\sim q_\theta(X_{t:T})}\left[\mathcal{S}(q_\theta(X_{t-1}|X_t))\right]\\
&-\mathcal{T}\cdot\sum_{t=1}^{T}\mathbb{E}_{X_{t-1:T}\sim q_\theta(X_{t-1:T})}\left[\log p(X_t|X_{t-1})\right]\\
&+\mathbb{E}_{X_{0:T}\sim q_\theta(X_{0:T})}\left[H(X_0)\right] + C,
\end{aligned} \qquad (2)$$

where $\mathcal{T}$ is the temperature, $C$ a parameter independent constant and $\mathcal{S}(.)$ the Shannon entropy. In practice, the expectation over $X_{0:T} \sim q_\theta(X_{0:T})$ is estimated using $M$ diffusion paths, where each diffusion path corresponds to a sample of $X_{0:T}$ from the model. The objective is optimized using the log-derivative trick to propagate the gradient through the expectation over $q_\theta$. Examination of Eq. 2 shows that the memory required for backpropagation scales linearly with the number of diffusion

steps, since backpropagation has to be performed through the expectation values for each time step $t$. Within a fixed memory budget, this results in a limitation on the number of diffusion steps and hence the model performance. To address these issues, we propose two alternatives to this objective, which are discussed in Sec. 3.

## 2.2 UNSUPERVISED COMBINATORIAL OPTIMIZATION

Sanokowski et al. (2024) apply diffusion models in UCO by reformulating it as an NPO problem. There is a wide class of CO problems that can be described in QUBO formulation (Lucas, 2014; Glover et al., 2022). In this case, the CO problem is described by an energy function $H_Q : \{0,1\}^N \to \mathbb{R}$ which is given by:

$$H_Q(X) = \sum_{i,j} Q_{ij} X_i X_j, \tag{3}$$

where $Q \in \mathbb{R}^{N \times N}$ is chosen according to the CO problem at hand. A table of the QUBO formulations of the CO problem types studied in this paper is given in Tab. 5. In UCO the goal is to train a conditional generative model $q_\theta(X|Q)$ on problem instances $Q$ that are drawn from a distribution $\mathcal{D}(Q)$ (see Sec. 5.1 and App. A.6) for more information on $\mathcal{D}(Q)$). After training the model can be used on unseen i.i.d CO problems to obtain solutions of high quality within a short amount of time. This can be realized by using the expectation of $H_Q(X)$ with respect to a parameterized probability distribution which is used as a loss function and minimized with respect to network parameters $\theta$:

$$L(\theta) = \mathbb{E}_{Q \sim \mathcal{D}(Q), X \sim q_\theta(X|Q)}[H_Q(X)]. \tag{4}$$

For notational convenience the conditional dependence of $q_\theta$ on the problem instance $Q$ is suppressed in the following. As minimizing the expectation value of $H_Q(X)$ in Eq. 4 is prone to getting stuck in local minima, numerous works (Hibat-Allah et al., 2021; Sun et al., 2022; Sanokowski et al., 2023; 2024) reframe this problem as an NPO problem and minimize $\mathcal{T} D_{KL}(q_\theta(X) \,||\, p_B(X)) = \mathbb{E}_{X \sim q_\theta(X)}[H_Q(X) + \mathcal{T} \log q_\theta(X)] + C$ instead, where $C$ is a constant which is independent of $\theta$. The optimization procedure of this objective is combined with annealing, where the objective is first optimized at high temperature, which is then gradually reduced to zero. At $\mathcal{T} = 0$ this objective reduces to the unconditional loss in Eq. 4. Sanokowski et al. (2023) motivate this so-called variational annealing procedure theoretically from a curriculum learning perspective and the aforementioned works show experimentally that it yields better solution qualities.

## 2.3 UNBIASED SAMPLING

When a parameterized probability distribution $q_\theta(X)$ is used to approximate the target distribution $p_B(X)$ the learned distribution will typically be an imperfect approximation. Consequently, samples from $q_\theta(X)$ will exhibit a bias. When the model is used to infer properties of the system that is described by the target distribution, it is essential to correct for this bias. The following paragraphs revisit two established unbiased sampling methods namely Self-Normalized Neural Importance Sampling (SN-NIS) and Neural Markov Chain Monte Carlo (NMCMC) that can be used to achieve this goal. These methods serve as the basis for our diffusion-based unbiased sampling methods which are introduced in Sec. 3.2.

**Self-Normalized Neural Importance Sampling:** SN-NIS allows asymptotically unbiased computation of expectation values of a target distribution. Given an observable $O : \{0,1\}^N \to \mathbb{R}$, an exact likelihood model $q_\theta(X)$ can be used to calculate expectation values $\langle O(X) \rangle_{p_B(X)} := \mathbb{E}_{p_B(X)}[O(X)]$ in the following way:

$$\langle O(X_0) \rangle_{p_B(X_0)} \approx \sum_{i=1}^{M} w(X^i) \, O(X^i),$$

where $X^i$ corresponds to the $i$-th of $M$ samples from $q_\theta(X)$. The importance weights are computed with $w(X^i) = \frac{\widehat{w}(X^i)}{\sum_j \widehat{w}(X^j)}$, where $\widehat{w}(X) = \frac{\widehat{p_B}(X)}{q_\theta(X)}$ (for a derivation we refer to App. A.2). The probability distribution that is proportional to $p_B(X) |O(X)|$ yields the minimum-variance estimate of $\langle O(X) \rangle_{p_B(X)}$ (Rubinstein & Kroese, 2016). However, in our experiments, we focus on a

distribution that approximates $p_B(X)$ since this allows the computation of expectations for various different $O$. An attractive feature of importance sampling is that it provides an unbiased estimator of the partition sum $\mathcal{Z}$ that is given by $\hat{\mathcal{Z}} = \frac{1}{M}\sum_{i=1}^{M} \widehat{w}(X^i)$. This estimator is used in the experiment section to estimate free energies (Sec. 5.2).

**Neural Markov Chain Monte Carlo:** NMCMC represents an alternative to SN-NIS which can be realized with the Metropolis-Hastings algorithm (Metropolis et al., 1953). Here, given a starting state $X$ a proposal state $X'$ is sampled from $q_\theta(X')$, which is accepted with the probability

$$A(X', X) = \min\left(1, \frac{\widehat{p}(X')q_\theta(X)}{\widehat{p}(X)q_\theta(X')}\right).$$

For more details on MCMC and Neural MCMC we refer to App. A.2.2. This process is repeated simultaneously for a batch of states until a convergence criterion is met (see App. A.3.4). After convergence the resulting samples can be considered to be approximately distributed as $X \sim p_B(X)$ and these samples can be used to estimate $\langle O(X) \rangle_{p_B(X)}$. Since diffusion models are approximate likelihood models, i.e. it is infeasible to compute $q_\theta(X)$ exactly, neither SN-NIS nor NMCMC is directly applicable to them. In Sec. 3.2 we propose techniques that overcome this limitation.

## 3 METHODS

### 3.1 SCALABLE DISCRETE DIFFUSION SAMPLERS

Sanokowski et al. (2024) demonstrate that increasing the number of diffusion steps in UCO improves the solution quality of the diffusion model, as it enables the model to represent more complex distributions. However, as discussed in Sec. 2.1, the loss function in Eq. 2 used in their work inflicts memory requirements that scale linearly with the number of diffusion steps. Given a fixed memory budget, this limitation severely restricts the expressivity of the diffusion model. In the following sections, we introduce training methods that mitigate this shortcoming.

**Forward KL Objective:** One possibility to mitigate the linear scaling issue is to use the forward Kullback-Leibler divergence (fKL). In contrast to the objective in Eq. 2 the gradient can be pulled into the expectation:

$$\nabla_\theta D_{KL}(p(X_{0:T})||q_\theta(X_{0:T})) = -\mathbb{E}_{X_{0:T}\sim p(X_{0:T})}[\nabla_\theta \log q_\theta(X_{0:T})].$$

However, since in NPO samples $X_{0:T} \sim p(X_{0:T})$ are not available, we employ SN-NIS to rewrite the expectation with respect to $X_{0:T} \sim q_\theta(X_{0:T})$. Note that this is feasible with diffusion models since they do provide exact joint likelihoods. In analogy to data-based diffusion models (Ho et al., 2020) one can now use Monte Carlo estimates of the sum over time steps $\log q_\theta(X_{0:T}) = \sum_{t=1}^{T} \log q_\theta(X_{t-1}|X_t)$ to mitigate the aforementioned memory scaling issue. The resulting gradient of the fKL objective is given by (see App. A.2.6):

$$\nabla_\theta D_{KL}(p(X_{0:T})||q_\theta(X_{0:T})) = -T\sum_{i=1}^{M}\mathbb{E}_{t\sim U\{1,...,T\}}\left[w(X_{0:T}^i)\nabla_\theta \log q_\theta(X_{t-1}^i|X_t^i)\right],$$

where $w(X_{0:T}^i) = \frac{\widehat{w}(X_{0:T}^i)}{\sum_{j=1}^{M}\widehat{w}(X_{0:T}^j)}$ are importance weights with $\widehat{w}(X_{0:T}^i) = \frac{\widehat{p}(X_{0:T}^i)}{q_\theta(X_{0:T}^i)}$, $X_{0:T}^i \sim q_\theta(X_{0:T})$, and $U\{1,...,T\}$ is the uniform distribution over the set $\{1,...,T\}$.

In the following, we will refer to this method as *SDDS: fKL w/ MC* since it realizes *Scalable Discrete Diffusion Samplers* (SDDS) using an objective that is based on the fKL, where the linear memory scaling issue is addressed with Monte Carlo estimation over diffusion steps. A pseudocode of the optimization procedure is given in App. A.3.5.

**Reverse KL Objective:** The minimization of the reverse Kullback-Leibler divergence (rKL) based objective function $L(\theta)$ introduced by Eq. 2 can be shown to be equivalent to parameter updates using the policy gradient theorem (Sutton & Barto, 2018) (see App. A.2.5). The resulting gradient updates are expressed as:

$$\nabla_\theta L(\theta) = -\mathbb{E}_{X_t\sim d^\theta(\mathcal{X},t), X_{t-1}\sim q_\theta(X_{t-1}|X_t)}\left[Q^\theta(X_{t-1}, X_t)\nabla_\theta \log q_\theta(X_{t-1}|X_t)\right], \quad (5)$$

where:

- $t = T$ in the first step and $t = 1$ is the terminal step,
- $Q^\theta(X_{t-1}, X_t) = R(X_t, X_{t-1}) + V^\theta(X_{t-1})$,
- $V^\theta(X_t) = \sum_{X_{t-1}} q_\theta(X_{t-1}|X_t) Q^\theta(X_{t-1}, X_t)$ where $V^\theta(X_0) = 0$,
- $R(X_t, X_{t-1})$ is defined as:

$$R(X_t, X_{t-1}) := \begin{cases} \mathcal{T}[\log p(X_t|X_{t-1}) - \log q_\theta(X_{t-1}|X_t)] & \text{if } 1 < t \leq T \\ \mathcal{T}[\log p(X_t|X_{t-1}) - \log q_\theta(X_{t-1}|X_t)] - H(X_{t-1}) & \text{if } t = 1. \end{cases}$$

Here $d^\theta(\mathcal{X}, t)$ represents the stationary state distribution of the state $(\mathcal{X}, t)$ and the policy $q_\theta$ in the setting of episodic RL environments.

This formulation suggests leveraging RL techniques to optimize Eq. 5, where $Q^\theta$ is the Q-function, $V^\theta$ the value function, and $R(X_t, X_{t-1})$ the reward. The usage of RL training methods addresses the linear memory scaling issue associated with Eq. 2 as sampling from the stationary state distribution $d^\theta$ corresponds in this setting to uniformly sampling diffusion time steps $t$. We chose to optimize Eq. 5 via the Proximal Policy Optimization (PPO) algorithm Schulman et al. (2017) (for details and pseudocode see App. A.3.6). In the following, we will refer to this method as *SDDS: rKL w/ RL* to emphasize that SDDSs are trained with the usage of RL methods.

## 3.2 Unbiased Sampling with Discrete Ising models

As concluded in Sec. 2.3, neither SN-NIS nor NMCMC can be applied with diffusion models. In the following, we introduce adapted versions each of these methods that allow us to perform unbiased sampling, i.e. unbiased computation of expectation values, with diffusion models.

**Self-Normalized Neural Importance Sampling for Diffusion Models:** Given a diffusion model $q_\theta$ that is trained to approximate a target distribution $p_B(X_0)$, we can use this model to calculate unbiased expectations $\langle O(X_0)\rangle_{p_B(X_0)}$ with SN-NIS in the following way (see App. A.2.1):

$$\langle O(X_0)\rangle_{p_B(X_0)} \approx \sum_{i=1}^{M} [w(X_{0:T}^i) O(X_0^i)]$$

where $w(X_{0:T}^i) = \frac{\widehat{w}(X_{0:T}^i)}{\sum_{j=1}^{M} \widehat{w}(X_{0:T}^j)}$ and $X_{0:T}^i \sim q_\theta(X_{0:T}^i)$ with $\widehat{w}(X_{0:T}^i) = \frac{\widehat{p}(X_{0:T}^i)}{q_\theta(X_{0:T}^i)}$. Using these importance weights the partition sum of $p_B(X_0)$ can be estimated with $\hat{\mathcal{Z}} = \frac{1}{M}\sum_{i=1}^{M} \widehat{w}(X_{0:T}^i)$.

**Neural MCMC for Diffusion Models:** Starting from an initial diffusion path $X_{0:T}$, we propose a state by sampling $X'_{0:T} \sim q(X'_{0:T})$. This diffusion path is then accepted with the probability (see App: A.2.4):

$$A(X', X) = \min\left(1, \frac{\widehat{p}(X'_{0:T}) q_\theta(X_{0:T})}{\widehat{p}(X_{0:T}) q_\theta(X'_{0:T})}\right)$$

This process is repeated until the Markov chain meets convergence criteria and samples $X_{0:T}$ are distributed as $p(X_{0:T})$ and $X_0$ can be considered to be distributed as $p_B(X_0)$. These samples can be used to approximate expectations with $\langle O(X_0)\rangle_{X_0 \sim p_B(X_0)}$ (see App. A.2.4).

## 4 Related Work

**Neural Optimization:** Besides their predominance in supervised and unsupervised learning tasks, neural networks become an increasingly popular choice for a wide range of data-free optimization tasks, i.e. scenarios where an objective function can be explicitly expressed rather than implicitly via data samples. In Physics Informed Neural Networks (Raissi et al., 2019) models are trained to represent the solutions of differential equations. Here the loss function measures the adherence of the solution quality. Similarly, Berzins et al. (2024) propose a neural optimization approach for generating shapes under geometric constraints. Recently, there has been increasing interest in using probabilistic generative models to generate solutions to neural optimization. Here the learned models do not directly represent a solution but rather a probability distribution over the solution space. We refer to this endeavor as Neural Probabilistic Optimization (NPO). In the following, we discuss two important NPO application areas in discrete domains.

**Neural Combinatorial Optimization:** Neural CO aims at generating high-quality solutions to CO problems time-efficiently during inference time. The goal is to train a generative model to generate solutions to a given CO problem instance on which it is conditioned. Supervised CO (Sun & Yang, 2023; Li et al., 2018; Böther et al., 2022a) typically involves training a conditional generative model using a training dataset that includes solutions obtained from classical solvers like Gurobi (Gurobi Optimization, LLC, 2023). However, as noted by Yehuda et al. (2020), these supervised approaches face challenges due to expensive data generation, leading to increased interest in unsupervised CO (UCO). In UCO the goal is to train models to solve CO problems without relying on labeled training data but only by evaluating the quality of generated solutions Bengio et al. (2021b). These methods often utilize exact likelihood models, such as mean-field models (Karalias & Loukas, 2020; Sun et al., 2022; Wang & Li, 2023). The calculation of expectation values in UCO is particularly convenient with mean-field models due to mathematical simplification arising from their assumption of statistical independence among modeled random variables. However, Sanokowski et al. (2023) demonstrate that the statistical independence assumption in mean-field models limits their performance on particularly challenging CO problems. They show that more expressive exact likelihood models, like autoregressive models, offer performance benefits, albeit at the cost of high memory requirements and longer sampling times, which slow down the training process. These limitations can be addressed by combining autoregressive models with RL methods to reduce memory requirements and accelerate training as it is done in Khalil et al. (2017) and Sanokowski et al. (2023). Sanokowski et al. (2023) additionally introduce Subgraph Tokenization to mitigate slow sampling and training in autoregressive models. Zhang et al. (2023) utilize GFlow networks (Bengio et al., 2021a), implementing autoregressive solution generation in UCO. Sanokowski et al. (2024) introduce a general framework that allows for the application of diffusion models to UCO and demonstrate their superiority on a range of popular CO benchmarks.

**Unbiased Sampling:** In this work, unbiased sampling refers to the task of calculating unbiased expectation values via samples from an approximation of the target distribution. Corresponding methods rely so far primarily on exact likelihood models, i.e. models that provide exact likelihoods for samples. Unbiased sampling plays a central role in a wide range of scientific fields, including molecular dynamics (Noé & Wu, 2018; Dibak et al., 2022), path tracing (Müller et al., 2019), and lattice gauge theory (Kanwar et al., 2020). These applications in continuous domains are suitable for using exact likelihood models like normalizing flows which are a popular model class in these domains. More recently approximate likelihood models became increasingly important in these applications since their increased expressivity yields superior results (Dibak et al., 2022; Zhang & Chen, 2022; Berner et al., 2022a; Jing et al., 2022; Berner et al., 2022b; Richter et al., 2023; Vargas et al., 2023; 2024; Akhound-Sadegh et al., 2024). In discrete domains, unbiased sampling arises as a key challenge in the study of spin glasses (Nicoli et al., 2020; McNaughton et al., 2020; Inack et al., 2022; Białas et al., 2022; Biazzo et al., 2024), many-body quantum physics (Sharir et al., 2020; Wu et al., 2021), and molecular biology (Cocco et al., 2018). In these settings, autoregressive models are the predominant model class. We are not aware of works that explore the applicability and performance of approximate likelihood models like diffusion models for unbiased sampling on discrete problem domains.

## 5 EXPERIMENTS

We evaluate our methods on UCO benchmarks in Sec. 5.1 and on two benchmarks for unbiased sampling Sec. 5.2 and in App. A.8.2. In all of our experiments, we use a time-conditioned diffusion model $q_\theta(X_{t-1}|X_t, t)$ that is realized either by a Graph Neural Network (GNN) (Scarselli et al., 2009) in UCO experiments or by a U-Net architecture (Ronneberger et al., 2015) in experiments on the Ising model (see App. A.4). In our experiments the probability distribution corresponding to individual reverse diffusion steps is parametrized via a product of Bernoulli dsitributions $q_\theta(X_{t-1}|X_t, t) = \prod_i^N \widehat{q}_\theta(X_t)_i^{X_{t-1,i}} (1 - \widehat{q}_\theta(X_t))_i^{1-X_{t-1,i}}$, where $\widehat{q}_\theta(X_t)_i := q_\theta(X_{t-1,i} = 1|X_t, t)$. As a noise distribution, we use the Bernoulli noise distribution from (Sohl-Dickstein et al., 2015) (see App. A.3.1).

### 5.1 UNSUPERVISED COMBINATORIAL OPTIMIZATION

In UCO the goal is to train a model to represent a distribution over solutions, which is conditioned on individual CO problem instances (see Sec. 2.2). Since each CO problem instance corresponds to a

graph it is a natural and popular choice to use GNNs for the conditioning on the CO problem instance (Cappart et al., 2021). Our experiments in UCO compare three objectives: the original DiffUCO objective as in Eq. 2 and the two newly proposed methods *SDDS: rKL w/ RL* and *SDDS: fKL w/ MC*. We evaluate these methods on benchmarks across four CO problem types: Maximum Independent Set (MIS), Maximum Clique (MaxCl), Minimum Dominating Set (MDS), and Maximum Cut (MC). For detailed explanations of these CO problem types see App. A.5. Following Zhang et al. (2023) and Sanokowski et al. (2024), we define the MIS and MaxCl problems on graphs generated by the RB-model (RB) which is known for producing particularly challenging problems (Xu et al., 2005). The MaxCut and MDS problem instances are defined on Barabasi-Albert (BA) graphs (Barabási & Albert, 1999). For each CO problem type except MaxCl, we evaluate the methods on both small and large graph datasets. The small datasets contain graphs with 200-300 nodes, while the large datasets have 800-1200 nodes. Each dataset comprises 4000 graphs for training, 500 for evaluation and 1000 for testing. To ensure a fair comparison in terms of available computational resources, we maintain a constant number of gradient update steps and a comparable training time across DiffUCO, *SDDS: rKL w/ RL* and *SDDS: fKL w/ MC* (see App. A.7.1). In our experiments, we first evaluate DiffUCO with fixed computational constraints. Using the same computational constraints, we then evaluate our proposed methods *SDDS: rKL w/ RL* and *SDDS: fKL w/ MC* with twice as many diffusion steps compared to DiffUCO. This is possible since these methods are designed to enable more diffusion steps with the same memory budget. Compared to the original DiffUCO implementation (DiffUCO (r), Sanokowski et al. (2024)) we also add a cosine learning rate schedule (Loshchilov & Hutter, 2017) and graph normalization layers (Cai et al., 2021) since this was found to improve the obtained results App. A.7.2. Additionally, the computational constraints between our DiffUCO evaluation and DiffUCO (r) are different. These two factors explain the superior performance of DiffUCO with respect to the reported values of DiffUCO (r) in Tab. 1 and Tab. 2. Sanokowski et al. (2024) have shown empirically that increasing the number of diffusion steps during inference improves the solution quality in UCO. In accordance with these insights, we evaluate the performance of the diffusion models with three times as many diffusion steps as during training.

**Results:** We report the average test dataset solution quality over 30 samples per CO problem instance. We include results for all three objectives and also include for reference the results from the two best-performing methods in DiffUCO (Sanokowski et al., 2024), and LTFT (Zhang et al., 2023). Results for the MIS and MDS problems are shown in Tab. 1 and for the MaxCl and MaxCut problems in Tab. 2. In these tables, we also show the solution quality of the classical method Gurobi, which is - if computationally feasible - run until the optimal solution is found. These results are intended to showcase the best possible achievable solution quality on these datasets. Since Gurobi runs on CPUs it cannot be compared straightforwardly to the other results which were obtained under specific constraints for GPUs. To ensure the feasibility of solutions and to obtain better samples from a product distribution in a deterministic way the final diffusion step is decoded with the Conditional Expectation (CE) (Raghavan, 1988) algorithm (see App: A.3.2). We optionally apply this method in the last diffusion step. However, we find that in our experiments the improvement by using Conditional Expectation (see App: A.3.2) is much smaller than the improvements reported in Sanokowski et al. (2024). We attribute this finding to the higher solution quality of our models. Secondly, we see that *SDDS: rKL w/ RL* outperforms all other methods in terms of average solution quality significantly in 4 out of 7 cases and insignificantly in 2 out of 7 cases. Only on MaxCut BA-large DiffUCO and *SDDS: fKL w/ MC* perform insignificantly better than *SDDS: rKL w/ RL*. In most cases, DiffUCO is the second best method and *SDDS: fKL w/ MC* performs worst. When increasing the number of samples from 30 to 150 sampled solutions (see Tab.3) *SDDS: fKL w/ MC* and *SDDS: rKL w/ RL* are the best-performing objectives in 6 of 7 cases and insignificantly the single best objective in 4 out of 7 cases. This finding is to be expected due to the mass-covering behavior of the fKL which allows the distribution $q_\theta$ to put probability mass on solutions where the target distribution has vanishing probability. As a result, the fKL-based training yields a worse average solution quality but due to the mass-covering property, the distribution covers more diverse solutions which makes it more likely that the very best solutions are within the support of $q_\theta$. In contrast to that, DiffUCO and *SDDS: rKL w/ RL* are mode seeking which tend to cover fewer solutions but exhibit a higher average solution quality. Our experimental results in Tab. 1 and Tab. 2 show consistent improvements over the results reported by Sanokowski et al. (2024).

| **MIS** | | RB-small | | RB-large | | **MDS** | | BA-small | | BA-large | |
|---|---|---|---|---|---|---|---|---|---|---|---|
| Method | Type | Size ↑ | time ↓ | Size ↑ | time ↓ | Method | Type | Size ↓ | time ↓ | Size ↓ | time ↓ |
| Gurobi | OR | $20.13 \pm 0.03$ | 6:29 | $42.51 \pm 0.06^*$ | 14:19:23 | Gurobi | OR | $27.84 \pm 0.00$ | 1:22 | $104.01 \pm 0.27$ | 3:35:15 |
| LTFT (r) | UL | 19.18 | 1:04 | 37.48 | 8:44 | LTFT (r) | UL | 28.61 | 4:16 | 110.28 | 1:04:24 |
| DiffUCO (r) | UL | $18.88 \pm 0.06$ | 0:14 | $38.10 \pm 0.13$ | 0:20 | DiffUCO (r) | UL | $28.30 \pm 0.10$ | 0:10 | $107.01 \pm 0.33$ | 0:10 |
| DiffUCO: CE (r) | UL | $19.24 \pm 0.05$ | 1:48 | $38.87 \pm 0.13$ | 9:54 | DiffUCO: CE (r) | UL | $28.20 \pm 0.09$ | 1:48 | $106.61 \pm 0.30$ | 6:56 |
| DiffUCO | UL | $19.42 \pm 0.03$ | 0:02 | $39.44 \pm 0.12$ | 0:03 | DiffUCO | UL | $28.10 \pm 0.01$ | 0:01 | $105.21 \pm 0.21$ | 0:01 |
| SDDS: *rKL w/ RL* | UL | $\mathbf{19.62 \pm 0.01}$ | 0:02 | $\mathbf{39.97 \pm 0.08}$ | 0:03 | SDDS: *rKL w/ RL* | UL | $\mathbf{28.03 \pm 0.00}$ | 0:02 | $\mathbf{105.16 \pm 0.21}$ | 0:02 |
| SDDS: *fKL w/ MC* | UL | $19.27 \pm 0.03$ | 0:02 | $38.44 \pm 0.06$ | 0:03 | SDDS: *fKL w/ MC* | UL | $28.34 \pm 0.02$ | 0:01 | $105.70 \pm 0.25$ | 0:02 |
| DiffUCO: CE | UL | $19.42 \pm 0.03$ | 0:20 | $39.49 \pm 0.09$ | 6:38 | DiffUCO: CE | UL | $28.09 \pm 0.01$ | 0:16 | $105.21 \pm 0.21$ | 1:45 |
| SDDS: *rKL w/ RL*-CE | UL | $\mathbf{19.62 \pm 0.01}$ | 0:20 | $\mathbf{39.99 \pm 0.08}$ | 6:35 | SDDS: *rKL w/ RL*-CE | UL | $\mathbf{28.02 \pm 0.01}$ | 0:16 | $\mathbf{105.15 \pm 0.20}$ | 1:41 |
| SDDS: *fKL w/ MC*-CE | UL | $19.27 \pm 0.03$ | 0:19 | $38.61 \pm 0.03$ | 6:31 | SDDS: *fKL w/ MC*-CE | UL | $28.33 \pm 0.02$ | 0:16 | $105.7 \pm 0.25$ | 1:41 |

Table 1: Left: Average independent set size on the test dataset of RB-small and RB-large. The higher the better. Right: Average dominating set size on the test dataset of BA-small and BA-large. The lower the set size the better. Left and Right: Total evaluation time is shown in h:m:s. (r) indicates that results are reported as in Sanokowski et al. (2024). ± represents the standard error over three independent training seeds. (CE) indicates that results are reported after applying conditional expectation. The best neural method is marked as bold. Gurobi results with * indicate that Gurobi was run with a time limit. On MIS RB-large the time-limit is set to 120 seconds per graph.

| **MaxCl** | | RB-small | | **MaxCut** | | BA-small | | BA-large | |
|---|---|---|---|---|---|---|---|---|---|
| Method | Type | Size ↑ | time ↓ | Method | Type | Size ↑ | time ↓ | Size ↑ | time ↓ |
| Gurobi | OR | $19.06 \pm 0.03$ | 11:00 | Gurobi (r) | OR | $730.87 \pm 2.35^*$ | 17:00:00 | $2944.38 \pm 0.86^*$ | 2:35:10:00 |
| LTFT (r) | UL | 16.24 | 1:24 | LTFT (r) | UL | 704 | 5:54 | 2864 | 42:40 |
| DiffUCO (r) | UL | $14.51 \pm 0.39$ | 0:08 | DiffUCO (r) | UL | $727.11 \pm 2.31$ | 0:08 | $2947.27 \pm 1.50$ | 0:08 |
| DiffUCO: CE (r) | UL | $16.22 \pm 0.09$ | 2:00 | DiffUCO: CE (r) | UL | $727.32 \pm 2.33$ | 2:00 | $2947.53 \pm 1.48$ | 7:34 |
| DiffUCO | UL | $17.40 \pm 0.02$ | 0:02 | DiffUCO | UL | $\mathbf{731.30 \pm 0.75}$ | 0:02 | $2974.60 \pm 7.73$ | 0:02 |
| SDDS: *rKL w/ RL* | UL | $\mathbf{18.89 \pm 0.04}$ | 0:02 | SDDS: *rKL w/ RL* | UL | $\mathbf{731.93 \pm 0.74}$ | 0:02 | $2971.62 \pm 8.15$ | 0:02 |
| SDDS: *fKL w/ MC* | UL | $18.40 \pm 0.02$ | 0:02 | SDDS: *fKL w/ MC* | UL | $\mathbf{731.48 \pm 0.69}$ | 0:02 | $2973.80 \pm 7.57$ | 0:02 |
| DiffUCO: CE | UL | $17.40 \pm 0.02$ | 0:38 | DiffUCO: CE | UL | $\mathbf{731.30 \pm 0.75}$ | 0:15 | $2974.64 \pm 7.74$ | 1:13 |
| SDDS: *rKL w/ RL*-CE | UL | $\mathbf{18.90 \pm 0.04}$ | 0:38 | SDDS: *rKL w/ RL*-CE | UL | $\mathbf{731.93 \pm 0.74}$ | 0:14 | $2971.62 \pm 8.15$ | 1:08 |
| SDDS: *fKL w/ MC*-CE | UL | $18.41 \pm 0.02$ | 0:38 | SDDS: *fKL w/ MC*-CE | UL | $\mathbf{731.48 \pm 0.69}$ | 0:14 | $2973.80 \pm 7.57$ | 1:08 |

Table 2: Left: Testset average clique size on the RB-small dataset. The larger the set size the better. Right: Average test set cut size on the BA-small and BA-large datasets. The larger the better. Left and Right: Total evaluation time is shown in d:h:m:s. (CE) indicates that results are reported after applying conditional expectation. Gurobi results with * indicate that Gurobi was run with a time limit. On MDS BA-small the time limit is set to 60 and on MDS BA-large to 300 seconds per graph.

## 5.2 UNBIASED SAMPLING OF ISING MODELS

In the discrete domain, the Ising model is frequently studied in the context of unbiased sampling (Nicoli et al., 2020; McNaughton et al., 2020). The Ising model is a discrete system, where the energy function $H_I : \{-1, 1\}^N \to \mathbb{R}$ is given by $H_I(\sigma) = -J \sum_{\langle i,j \rangle} \sigma_i \sigma_j$, where $\langle i, j \rangle$ runs over all neighboring pairs on a lattice. At temperature $\mathcal{T}$ the state of the system in thermal equilibrium is described by the Boltzmann distribution from Eq. 1. Analogously to Nicoli et al. (2020), we explore unbiased sampling using finite-size Ising models (see Sec. 2.3) on a periodic, regular 2D grid of size $L$ with a nearest-neighbor coupling parameter of $J = 1$. We experimentally validate our unbiased sampling approach with diffusion models by comparing the estimated values of the free energy $\mathcal{F} = -\frac{1}{\beta} \log \mathcal{Z}$, internal energy $\mathcal{U} = \sum_X p_B(X) H_I(X)$, and entropy $\mathcal{S} = \beta (\mathcal{U} - \mathcal{F})$ against the theoretical values derived by Ferdinand & Fisher (1969). We also report the effective sample size per sample $\epsilon_{\text{eff}}/M := \frac{1}{M} \frac{(\sum_{i=1}^M w_i)^2}{\sum_{i=1}^M w_i^2}$ of each method. For the best possible model, the effective sample size per sample equals one as $w_i = 1/M \; \forall \; i \in \{1, ..., M\}$. For the worst possible model $\epsilon_{\text{eff}}/M = 1/M$ as there is one weight which is equal to 1 and all others are 0. In our experiments, we train diffusion models using 300 diffusion steps and a U-net architecture (details in App. A.4 and App. A.7.3). Tab. 4 presents our results where each model is evaluated over three independent sampling seeds. We compare our methods to two other methods that both rely on the rKL objective. First, the AR models by (Wu et al., 2019) which we label as VAN (r), and second the NIS method with AR models by (Nicoli et al., 2020) which we label as AR (r). We also evaluate an AR reimplementation of (Nicoli et al., 2020) using the same architecture and computational constraints as the diffusion models.

**Results:** We find that the diffusion model outperforms the AR baseline reported in (Nicoli et al., 2020). Our method *SDDS: fKL w/ MC* yields the best performance, producing values closest to

| CO problem type | MaxCl ↑ | MaxCut ↑ | | MIS ↑ | | MDS ↓ | |
|---|---|---|---|---|---|---|---|
| Graph Dataset | RB-small | BA-small | BA-large | BA-small | BA-large | BA-small | BA-large |
| Gurobi Set Size | $19.06 \pm 0.03$ | $730.87 \pm 2.35^*$ | $2944.38 \pm 0.86^*$ | $20.14 \pm 0.04$ | $42.51 \pm 0.06^*$ | $27.81 \pm 0.08$ | $104.01 \pm 0.27$ |
| DiffUCO: CE | $18.34 \pm 0.07$ | $\mathbf{732.64 \pm 0.74}$ | $\mathbf{2979.09 \pm 6.69}$ | $19.79 \pm 0.04$ | $41.84 \pm 0.07$ | $\mathbf{27.97 \pm 0.02}$ | $104.36 \pm 0.22$ |
| SDDS: *rKL w/ RL*-CE | $19.05 \pm 0.03$ | $732.78 \pm 0.74$ | $2979.05 \pm 6.69$ | $20.02 \pm 0.02$ | $42.12 \pm 0.06$ | $27.89 \pm 0.01$ | $104.26 \pm 0.21$ |
| SDDS: *fKL w/ MC*-CE | $\mathbf{19.06 \pm 0.04}$ | $\mathbf{733.06 \pm 0.69}$ | $\mathbf{2979.88 \pm 6.65}$ | $\mathbf{20.05 \pm 0.03}$ | $41.23 \pm 0.05$ | $27.89 \pm 0.01$ | $104.36 \pm 0.23$ |

Table 3: Comparison of the best solution quality out of 150 samples for each CO problem instance averaged over the test dataset. Arrows $\downarrow, \uparrow$ indicate whether higher or lower is better. Gurobi results with $^*$ indicate that Gurobi was run with a time limit (see Tab. 1 and Tab. 2).

| $24 \times 24$ grid | Free Energy $\mathcal{F}/\mathbf{L^2}$ | Internal Energy $\mathcal{U}/\mathbf{L^2}$ | | Entropy $\mathcal{S}/\mathbf{L^2}$ | $\epsilon_{\text{eff}}/\mathbf{M}$ |
|---|---|---|---|---|---|
| Optimal value | $-2.11215$ | $-1.44025$ | | $0.29611$ | $1$ |
| VAN (r) (Nicoli et al., 2020) | $-2.10715 \pm 0.0000$ | $-1.5058 \pm 0.0001$ | N/A | $0.26505 \pm 0.00004$ | N/A |
| Unb. sampling Methods | SN-NIS | SN-NIS | NMCMC | SN-NIS | – |
| AR (r) (Nicoli et al., 2020) | $-2.1128 \pm 0.0008$ | $-1.43 \pm 0.02$ | $-1.448 \pm 0.007$ | $0.299 \pm 0.007$ | N/A |
| AR | $-2.09344 \pm 0.00063$ | $-1.65420 \pm 0.00562$ | $-1.68479 \pm 0.00198$ | $0.19357 \pm 0.002$ | $0.00006 \pm 0.00002$ |
| SDDS: *rKL w/ RL* | $-2.11150 \pm 0.00062$ | $-1.44910 \pm 0.01412$ | $-1.45225 \pm 0.00152$ | $0.29192 \pm 0.00615$ | $0.00023 \pm 0.00017$ |
| SDDS: *fKL w/ MC* | $\mathbf{-2.11209 \pm 0.00008}$ | $\mathbf{-1.4410 \pm 0.0008}$ | $\mathbf{-1.44264 \pm 0.00187}$ | $\mathbf{0.29573 \pm 0.0004}$ | $\mathbf{0.0102 \pm 0.0024}$ |

Table 4: Comparison of estimated observables $\mathcal{F}/L^2, \mathcal{U}/L^2, \mathcal{S}/L^2$ and the effective sample size per sample $\epsilon_{\text{eff}}/M$ of an Ising model of size $24 \times 24$ at critical inverse temperature $\beta_c = 0.4407$ using SN-NIS and NMCMC methods.

theoretical predictions. This aligns with expectations due to fKL's mass-covering property, which should improve the model's coverage of the target distribution which is beneficial in unbiased sampling. The diffusion model trained with *SDDS: rKL w/ RL* performs better than our AR reimplementation but worse than both the *SDDS: fKL w/ MC* model and the reported AR baseline (AR (r)). We do not report the performance of DiffUCO as this method suffered from severe mode-seeking behavior in this setting, where it only predicted either $+1$ or $-1$ for every state variable, resulting in poor behavior for unbiased sampling. Our experiments demonstrate that discrete diffusion models offer a promising alternative to AR models for unbiased sampling. Key advantages of diffusion models include flexibility in the number of diffusion steps and forward passes, which can be adjusted as a hyperparameter. In contrast, the number of forward passes in AR models is fixed to the dimension of the Ising model state. Our diffusion models achieve better performance while using only 300 diffusion steps, which are significantly fewer than the 576 forward passes required by the AR baseline. Since (Nicoli et al., 2020) ran their experiments on different types of GPUs it is in principle not possible to deduce from these results that AR models are inferior to diffusion models. Therefore, we complement our experimental evaluation with our own implementation of an AR approach. We utilize the same U-net architecture as for our diffusion models and the same computational resources. The corresponding results (AR in Tab. 4) indicate that also under these conditions the diffusion model approaches excel.

## 6   LIMITATIONS AND CONCLUSION

This work introduces *Scalable Discrete Diffusion Samplers* (SDDS) based on novel training methods that enable the implementation of discrete diffusion models with an increased number of diffusion steps in Unsupervised Combinatorial Optimization (UCO) and unbiased sampling problems. We demonstrate that the reverse KL objective of discrete diffusion samplers can be optimized efficiently using Reinforcement Learning (RL) methods. Additionally, we introduce an alternative training method based on Self-Normalized Importance Sampling of the gradients of the forward KL divergence. Both methods facilitate mini-batching across diffusion steps, allowing for more diffusion steps with a given memory budget. Our methods achieve state-of-the-art on popular challenging UCO benchmarks. For unbiased sampling in discrete domains, we extend existing importance sampling and Markov Chain Monte Carlo methods to be applicable to diffusion models. Furthermore, we show that discrete diffusion models can outperform popular autoregressive approaches in estimating observables of discrete distributions. Future research directions include leveraging recent advances in discrete score matching (Lou et al., 2024) to potentially improve the performance of SN-NIS-based objectives in UCO. While the reverse KL-based objective introduces new optimization hyperparameters, our experiments suggest that these require minimal fine-tuning (see App. A.10). Overall, SDDS represents a principled and efficient framework for leveraging diffusion models in discrete optimization and sampling tasks.

ACKNOWLEDGMENTS

The ELLIS Unit Linz, the LIT AI Lab, the Institute for Machine Learning, are supported by the Federal State Upper Austria. We thank the projects INCONTROL-RL (FFG-881064), PRI-MAL (FFG-873979), S3AI (FFG-872172), DL for GranularFlow (FFG-871302), EPILEPSIA (FFG892171), FWF AIRI FG 9-N (10.55776/FG9), AI4GreenHeatingGrids (FFG-899943), INTE-GRATE (FFG-892418), ELISE (H2020-ICT-2019-3 ID: 951847), Stars4Waters (HORIZON-CL6-2021- CLIMATE-01-01). We thank NXAI GmbH, Audi.JKU Deep Learning Center, TGW LO-GISTICS GROUP GMBH, Silicon Austria Labs (SAL), FILL Gesellschaft mbH, Anyline GmbH, Google, ZF Friedrichshafen AG, Robert Bosch GmbH, UCB Biopharma SRL, Merck Healthcare KGaA, Verbund AG, GLS (Univ. Waterloo), Software Competence Center Hagenberg GmbH, Bo-realis AG, TÜV Austria, Frauscher Sensonic, TRUMPF and the NVIDIA Corporation. We ac-knowledge EuroHPC Joint Undertaking for awarding us access to Meluxina, Vega, and Karolina at IT4Innovations.

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

# A APPENDIX

## A.1 DERIVATIONS

### A.1.1 IMPORTANCE SAMPLING AND NEURAL IMPORTANCE SAMPLING

Importance Sampling (IS) is a well-established Monte Carlo method used to estimate expectations of observables $O(X)$ under a target distribution $p(X)$ when direct sampling from $p$ is challenging. The core idea is to use a proposal distribution $q(X)$ which is easy to sample from and proposes samples where $p(X)$ or ideally $p(X)\,|O(X)|$ is large. IS can be used to calculate the expectation of an observable $O(X)$ in the following way:

$$O(X) = \sum_X p(X)\,O(X) = \sum_X q(X)\frac{p(X)}{q(X)}O(X) = \mathbb{E}_{X \sim q(X)}\left[\frac{p(X)}{q(X)}O(X)\right] \tag{6}$$

However, this approach makes it necessary to design a suitable proposal distribution $q(X)$, which is not possible in many cases. Therefore, Neural Importance Sampling can be used instead, where a distribution $q_\theta(X)$ is parameterized using a Neural Network and trained to approximate the target distribution. By replacing $q(X)$ in Eq. 6 with $q_\theta(X)$ the Neural Importance Sampling estimator is then given by:

$$O(X) = \mathbb{E}_{X \sim q_\theta(X)}\left[\frac{p(X)}{q_\theta(X)}O(X)\right]$$

## A.2 SELF-NORMALIZED NEURAL IMPORTANCE SAMPLING

In some cases, when an unnormalized target distribution is given, i.e. it is infeasible to calculate the normalization constant $\mathcal{Z}$, IS or NIS cannot straightforwardly be applied, as this requires the computation of $p(X)$ and therefore $\mathcal{Z}$. To mitigate this issue, Self-Normalized Importance Sampling (SNIS) can be employed Rubinstein & Kroese (2016). The estimator is given by:

$$\mathbb{E}_{p(X)}[O(X)] = \sum_X p(X)O(X) \approx \sum_{i=1}^N w(X_i)O(X_i),$$

where $w(X_i) = \frac{\hat{w}(X_i)}{\sum_j \hat{w}(X_j)}$ with $\hat{w}(X_i) = \frac{p(X_i)}{q(X_i)}$ are the importance weights, and $X_i \sim q(X)$.

**Derivation:** When $p(X)$ is unnormalized, i.e., $p(X) = \tilde{p}(X)/\mathcal{Z}$, where $\tilde{p}(X)$ is the unnormalized distribution and $\mathcal{Z}$ is the unknown normalization constant, the weights $\tilde{w}(X_i) = \frac{\tilde{p}(X_i)}{\mathcal{Z}q(X_i)}$ depend on $\mathcal{Z}$, which cannot be computed. To circumvent this, Self-Normalized Importance Sampling redefines the weights as normalized importance weights:

$$\hat{w} = \frac{\tilde{p}(X_i)}{q(X_i)}, \quad w(X_i) = \frac{\tilde{w}(X_i)}{\sum_{j=1}^N \tilde{w}(X_j)} = \frac{\hat{w}(X_i)}{\sum_{j=1}^N \hat{w}(X_j)}.$$

Using these normalized weights, the expectation can be estimated as:

$$\mathbb{E}_{p(X)}[O(X)] = \sum_X q(X)\tilde{w}(X)O(X) = \frac{\sum_X q(X)\tilde{w}(X)O(X)}{\sum_X q(X)\tilde{w}(X)}$$

$$= \frac{\mathbb{E}_{X \sim q(X)}[\tilde{w}(X)O(X)]}{\mathbb{E}_{X \sim q(X)}[\tilde{w}(X)]} \approx \frac{\sum_{i=0}^N \hat{w}(X_i)O(X_i)}{\sum_{j=1}^N \hat{w}(X_j)} = \sum_{i=1}^N w(X_i)O(X_i),$$

This approach avoids the need to compute $\mathcal{Z}$ explicitly, as the normalization is handled by the sum of the unnormalized weights $\tilde{w}(X_i)$. In practice, this is particularly useful for unnormalized target distributions or when $\mathcal{Z}$ is computationally expensive to estimate.

When using Neural Importance Sampling, the proposal distribution $q(X)$ is replaced with a parameterized distribution $q_\theta(X)$, and the SNIS estimator becomes:

$$\mathbb{E}_{p(X)}[O(X)] \approx \frac{\sum_{i=1}^N \frac{\tilde{p}(X_i)}{q_\theta(X_i)} O(X_i)}{\sum_{j=1}^N \frac{\tilde{p}(X_j)}{q_\theta(X_j)}}.$$

### A.2.1 Neural Importance Sampling with Diffusion Models

In the following, it will be shown that the expectation of an observable $O : \{0,1\}^N \to \mathbb{R}$ can be computed with:

$$\langle O(X_0)\rangle_{p_B(X_0)} \approx \sum_i [w(X_{0:T,i}) O(X_{0,i})] \tag{7}$$

where $w(X_{i,0:T}) = \frac{\widehat{w}(X_{i,0:T})}{\sum_j \widehat{w}(X_{j,0:T})}$ and $X_{i,0:T} \sim q_\theta(X_{0:T})$ with $\widehat{w}(X_{i,0:T}) = \frac{\widehat{p}(X_{i,0:T})}{q_\theta(X_{i,0:T})}$.

To show that we start with

$$\langle O(X_0)\rangle_{p_B(X_0)} := \sum_{X_0} p_B(X_0)\, O(X_0) = \sum_{X_{0:T}} p(X_{0:T}) O(X_0) \tag{8}$$

where we introduce new variables $X_{1:T}$ which are distributed according to the distribution $p(X_{1:T}|X_0)$. We then used that $p(X_{0:T}) = p(X_{1:T}|X_0)\, p_B(X_0)$ and that $\sum_{X_{1:T}} p(X_{1:T}|X_0)\, p_B(X_0) = p_B(X_0)$. Finally, we estimate the right hand side of Eq. 8 with Neural Importance Sampling by inserting $1 = \frac{q_\theta(X_{0:T})}{q_\theta(X_{0:T})}$ to arrive at

$$\langle O(X_0)\rangle_{p_B(X_0)} = \mathbb{E}_{X_{0:T} \sim q_\theta(X_{0:T})}\left[\frac{p(X_{0:T})}{q_\theta(X_{0:T})} O(X_0)\right].$$

As $p_B(X_0)$ is only known up to its normalization constant $\mathcal{Z}$ we employ SN-NIS (see Sec. 2.3).

### A.2.2 MCMC

The Metropolis-Hastings algorithm (Metropolis et al., 1953) is a standard method to obtain unbiased samples from a target distribution $p(X)$. Starting from an initial state $X$, a proposal state $X'$ is accepted with the acceptance probability of

$$A(X', X) = \min\left(1, \frac{\omega(X|X')\, \widehat{p}(X')}{\omega(X'|X)\, \widehat{p}(X)}\right),$$

where $\omega(X|X')$ is the transition probability from $X'$ to $X$ and is often chosen so that it is symmetric and satisfies $\omega(X|X') = \omega(X'|X)$. Here, $A(X', X)$ is chosen in a way so that the detailed balance condition $A(X, X')\, \omega(X|X')\, p(X) = A(X', X)\, \omega(X'|X)\, p(X')$ is satisfied.

### A.2.3 Neural MCMC

This acceptance probability can be adapted to Neural MCMC by replacing $\omega(X|X')$ with a probability distribution that is parameterized by a neural network $q_\theta(X)$, which approximates the target distribution (Nicoli et al., 2020). The acceptance probability is then given by:

$$A(X', X) = \min\left(1, \frac{q_\theta(X)\, \widehat{p}(X')}{q_\theta(X')\, \widehat{p}(X)}\right).$$

However, for diffusion models $q_\theta(X)$ is intractable and the formulation above can therefore not be applied to diffusion models. We will derive Neural MCMC for diffusion models in the following section.

### A.2.4 NEURAL MCMC WITH DIFFUSION MODELS

We adapt NMCMC to diffusion models to obtain trajectories that are approximately distributed according to the target distribution $p(X_{0:T})$ and show that these samples can be used to compute $\langle O(X_0) \rangle_{X_0 \sim p_B(X_0)}$.

This process is usually repeated until a convergence criterion is met. The resulting final state is approximately distributed according to the target distribution $p$. In NMCMC $\omega(X|X')$ is set to the approximating distribution $q_\theta(X)$, so that the acceptance probability is given by

$$A(X'_{0:T}, X_{0:T}) = \min\left(1, \frac{q_\theta(X_{0:T})\,\widehat{p}(X'_{0:T})}{q_\theta(X'_{0:T})\,\widehat{p}(X_{0:T})}\right),$$

where $Y$ is substituted with $X_{0:T}$ and $Y'$ with $X'_{0:T}$. Thus it becomes apparent that these updates can be used to obtain unbiased diffusion paths $X_{0:T} \sim p(X_{0:T})$ of which $X_0 \sim p_B(X_0)$.

The resulting diffusion paths $X_{0:T}$ are then distributed as $p(X_{0:T})$ and samples $X_0$ can then be used to calculate expectations $\langle O(X_0) \rangle_{X_0 \sim p_B(X_0)}$.

*Proof.* The statement follows from

$$\begin{aligned}
\langle O(X_0) \rangle_{p(X_{0:T})} &= \sum_{X_{0:T}} p(X_{0:T})O(X_0) \\
&= \sum_{X_0} p_B(X_0)O(X_0) = \langle O(X_0) \rangle_{p_B(X_0)},
\end{aligned}$$

where we have used that $O(X_0)$ does not depend on $X_{1:T}$ which is why $\sum_{X_{1:T}} p(X_{1:T}) = 1$. $\quad\square$

### A.2.5 POLICY GRADIENT THEOREM FOR DATA PROCESSING INEQUALITY

To prove that the Data Processing Inequality (see Sec. 3.1) can be optimized with the usage of RL we first define

$$V^\theta(X_t) = \sum_{X_{t-1}} q_\theta(X_{t-1}|X_t)\,Q^\theta(X_{t-1}, X_t) \tag{9}$$

where

- $t = T$ in the first step
- $t = 1$ is the terminal step
- $Q^\theta(X_{t-1}, X_t) = R(X_t, X_{t-1}) + V^\theta(X_{t-1})$
- $V^\theta(X_t) = \sum_{X_{t-1}} q_\theta(X_{t-1}|X_t)Q^\theta(X_{t-1}, X_t)$ where $V^\theta(X_0) = 0$
- $R(X_t, X_{t-1})$ is defined as:

$$R(X_t, X_{t-1}) := \begin{cases} \mathcal{T}[\log p(X_t|X_{t-1}) - \log q_\theta(X_{t-1}|X_t)] & \text{if } 1 < t \leq T \\ \mathcal{T}[\log p(X_t|X_{t-1}) - \log q_\theta(X_{t-1}|X_t)] - H(X_{t-1}) & \text{if } t = 1 \end{cases}$$

Then with the recursive application of Eq. 9 on $-\mathbb{E}_{X_T \sim q(X_T)}[V^\theta(X_T)]$ it can be shown that

$$\begin{aligned}
-\mathbb{E}_{X_T \sim q(X_T)}[V^\theta(X_T)] &= \mathcal{T} \cdot \sum_{t=1}^{T} \mathbb{E}_{X_{t-1:T} \sim q_\theta(X_{t-1:T})}\left[\log q_\theta(X_{t-1}|X_t)\right] \\
&\quad - \mathcal{T} \cdot \sum_{t=1}^{T} \mathbb{E}_{X_{t-1:T} \sim q_\theta(X_{t-1:T})}\left[\log p(X_t|X_{t-1})\right] \\
&\quad + \mathbb{E}_{X_{0:T} \sim q_\theta(X_{0:T})}\left[H(X_0)\right] \\
&= \mathcal{T}\,D_{KL}(q_\theta(X_{0:T})||p(X_{0:T})) + \widetilde{C}
\end{aligned} \tag{10}$$

Where $\widehat{C} = -\mathcal{T}\,\mathbb{E}_{X_T \sim q(X_T)}[\log q(X_T)] - \mathcal{T}\log \mathcal{Z}$

*Proof.* In the following, we will prove the first equality of Eq. 10 by induction. First show that this equality holds for $T = 1$:

$$-\mathbb{E}_{X_1 \sim q(X_1)}[V^\theta(X_1)] = -\mathbb{E}_{X_1 \sim q(X_1)}[\sum_{X_0} q_\theta(X_0|X_1)Q^\theta(X_0, X_1)]$$

$$= -\mathbb{E}_{X_1 \sim q(X_1)}[\sum_{X_0} q_\theta(X_0|X_1)[R(X_1, X_0) + V^\theta(X_0)]]$$

$$= -\mathbb{E}_{X_1 \sim q(X_1)}[\sum_{X_0} q_\theta(X_0|X_1) [\mathcal{T}[\log p(X_1|X_0) - \log q_\theta(X_0|X_1)] - H(X_0)]$$

$$= \mathcal{T} \cdot \mathbb{E}_{X_{0:1} \sim q_\theta(X_{0:1})} [\log q_\theta(X_0|X_1)] - \mathcal{T} \cdot \mathbb{E}_{X_{0:1} \sim q_\theta(X_{0:1})} [\log p(X_1|X_0)]$$
$$+ \mathbb{E}_{X_{0:1} \sim q_\theta(X_{0:1})} [H(X_0)]$$

Where it is apparent that this expression is equal to the right-hand side of Eq. 10 when $T = 1$.

Next, we have to show that assuming it holds for $T$ it also holds for $T + 1$.

$$-\mathbb{E}_{X_{T+1} \sim q(X_{T+1})}[V^\theta(X_{T+1})] = -\mathbb{E}_{X_{T+1} \sim q(X_{T+1})}[\sum_{X_T} q_\theta(X_T|X_{T+1})[R(X_{T+1}, X_T) + V^\theta(X_T)]]$$

$$= \mathbb{E}_{X_{T+1} \sim q(X_{T+1})}[\sum_{X_T} q_\theta(X_T|X_{T+1})[\mathcal{T}[\log p(X_{T+1}|X_T)$$
$$- \log q_\theta(X_T|X_{T+1})] + V^\theta(X_T)]]$$

$$= \mathcal{T} \cdot \sum_{t=1}^{T+1} \mathbb{E}_{X_{T+1:t-1} \sim q_\theta(X_{T+1:t-1})} [\log q_\theta(X_{t-1}|X_t)]$$

$$- \mathcal{T} \cdot \sum_{t=1}^{T+1} \mathbb{E}_{X_{T+1:t-1} \sim q_\theta(X_{T+1:t-1})} [\log p(X_t|X_{t-1})]$$

$$+ \mathbb{E}_{X_{T+1:0} \sim q_\theta(X_{T+1:0})} [H(X_0)]$$

Therefore we have proven the statement by induction. $\square$

As we have shown that the objective $\mathcal{T} D_{KL}(q_\theta(X_{0:T})||p(X_{0:T}))$ can be minimized by minimizing $\mathbb{E}_{X_T \sim q(X_T)}[V^\theta(X_T)]$. Applying the Policy Gradient Theorem for episodic Markov Decision Processes (MDP) (Sutton & Barto (2018); Sec. 13.2), it can be shown that

$$\nabla_\theta L(\theta) = -\nabla_\theta \mathbb{E}_{X_T \sim q(X_T)}[V^\theta(X_T)]$$
$$= -\mathbb{E}_{X_t \sim d^\theta(\mathcal{X}, t), X_{t-1} \sim q_\theta(X_{t-1}|X_t)} \left[ Q^\theta(X_{t-1}, X_t) \nabla_\theta \log q_\theta(X_{t-1}|X_t) \right],$$

where $d^\theta(\mathcal{X}, t)$ is the stationary distribution for $q_\theta$ under the episodic MDP. We use the PPO algorithm to minimize this objective as explained in App. A.3.6.

Usually, the reward is not allowed to depend on network parameters $\theta$, but the entropy regularization in the form of $\mathcal{T} \log q_\theta(X)$ is an exception due to the property that $\nabla_\theta \mathbb{E}_{X \sim q_\theta(X)} [\log q_\theta(X)] = \mathbb{E}_{X \sim q_\theta(X)} [\nabla_\theta \log q_\theta(X)] = \sum_X q_\theta(X) \frac{1}{q_\theta(X)} \nabla_\theta q_\theta(X) = \nabla_\theta \sum_X q_\theta(X) = \nabla_\theta 1 = 0$.

### A.2.6 NEURAL IMPORTANCE SAMPLING GRADIENT OF FORWARD KL DIVERGENCE

In the following, we will show that the gradient of the fKL between the forward and reverse diffusion path can be approximated with:

$$\nabla_\theta D_{KL}(p(X_{0:T})||q_\theta(X_{0:T})) = -T \sum_i \mathbb{E}_{t \sim U\{1,...,T\}} \left[ w(X_{0:T}^i) \nabla_\theta \log q_\theta(X_{t-1}^i|X_t^i) \right],$$

where $w(X_{0:T}^i) = \frac{\widehat{w}(X_{0:T}^i)}{\sum_j \widehat{w}(X_{0:T}^j)}$ and $X_{0:T}^i \sim q_\theta(X_{0:T})$ with $\widehat{w}(X_{0:T}^i) = \frac{\widehat{p}(X_{0:T}^i)}{q_\theta(X_{0:T}^i)}$.

This follows from

$$
\begin{aligned}
\nabla_\theta D_{KL}(p(X_{0:T})||q_\theta(X_{0:T})) &= -\mathbb{E}_{X_{0:T}\sim p(X_{0:T})}[\nabla_\theta \log q_\theta(X_{0:T})] \\
&= -\mathbb{E}_{X_{0:T}\sim q_\theta(X_{0:T})}\left[\frac{p(X_{0:T})}{q_\theta(X_{0:T})}\nabla_\theta \log q_\theta(X_{0:T})\right] \\
&= -T\,\mathbb{E}_{X_{0:T}\sim q_\theta(X_{0:T}),t\sim U(\{1,...,T\})}\left[\frac{p(X_{0:T})}{q_\theta(X_{0:T})}\nabla_\theta \log q_\theta(X_{t-1}|X_t)\right] \\
&= -T\sum_i \mathbb{E}_{t\sim U\{1,...,T\}}\left[w(X_{0:T}^i)\nabla_\theta \log q_\theta(X_{t-1}^i|X_t^i)\right],
\end{aligned}
$$

where we have used in the first equality that $\log p(X_{0:T})$ does not depend on network parameters. In the second equality we have insert $1 = \frac{q_\theta(X_{0:T})}{q_\theta(X_{0:T})}$ and rewrite the expectation over $q_\theta(X_{0:T})$. In the third equality we use that $\log q_\theta(X_{0:T}) = \sum_{t=0}^T \log q_\theta(X_{t-1}|X_t)$. We can then make a Monte Carlo estimate of this sum with $\log q_\theta(X_{0:T}) = T\,\mathbb{E}_{t\sim U\{0,...,T\}}[\log q_\theta(X_{t-1}|X_t)]$. In the fourth equality we apply SN-NIS so that the partition sum in $p(X_{0:T})$ cancels out.

## A.3 ALGORITHMS

### A.3.1 NOISE DISTRIBUTION

The Bernoulli Noise Distribution is given by:

$$
p(X_{t,i}|X_{t-1}) = \begin{cases} (1-\beta_t)^{1-X_{t-1,i}} \cdot \beta_t^{X_{t-1,i}} & \text{for } X_{t,i} = 0 \\ (1-\beta_t)^{X_{t-1,i}} \cdot \beta_t^{1-X_{t-1,i}} & \text{for } X_{t,i} = 1, \end{cases}
$$

where $\beta_t$ is the noise parameter. Sanokowski et al. (2024) use a noise schedule of $\beta_t = \frac{1}{T-t+2}$. However, we instead use an exponential noise schedule which is given by $\beta_t = \frac{1}{2}\exp\left(-k\left(1-\frac{t}{T}\right)\right)$ with $k = 6\log(2)$. Our experiments are always conducted with this schedule.

### A.3.2 CONDITIONAL EXPECTATION

Conditional Expectation (CE) is an iterative method for sampling from a product $p(X) = \prod_i p(X_i)$ distribution to obtain solutions of above-average quality Raghavan (1988); Karalias & Loukas (2020). We define a vector $v$ of Bernoulli probabilities, where each component $v_i = p(X_i)$. The CE process involves these steps:

1. Sort the components of $v$ in descending order to obtain a sorted probability vector $p$
2. Starting with $i = 0$, create two vectors:
   - $\omega_0$: Set the $i$-th component to 0
   - $\omega_1$: Set the $i$-th component to 1

   Initially, $\omega_0 = (0, p_1, \ldots, p_N)$ and $\omega_1 = (1, p_1, \ldots, p_N)$.
3. Compute $H(\omega_0)$ and $H(\omega_1)$.
4. Update $v$ to the configuration $\omega_j$, where $j = \arg\min_{l\in\{0,1\}} H(\omega_l)$.
5. Increment $i$ to $i + 1$.
6. Repeat steps 2-5 until all $v_i$ are either 0 or 1.

This process progressively yields better-than-average values for each component of $v$. With the choice of energy functions taken in App A.5 CE always removes constraint violations from generated solutions. In our experiments, we speed up the CE inference time by a large factor by providing a fast GPU implementation that leverages jax.lax.scan (see time column in results denoted with *-CE* in Tab. 2 and Tab. 1).

### A.3.3 ASYMPTOTICALLY UNBIASED SAMPLING

**Autoregressive Asymptotically Unbiased Sampling:**

While SN-NIS and NMCMC can be used to remove some of the bias, the bias cannot be completely removed when the model suffers from a lack of coverage, i.e. $\exists X$ such that $q_\theta(X) = 0$ and $p_B(X)O(x) \neq 0$ (Owen, 2013). A way to mitigate this issue is to adapt the model so that $q_\theta(X) > 0 \; \forall \; X$. In autoregressive models $q_\theta(X) = \prod_i^N q_\theta(X_i|X_{<i})$ Nicoli et al. (2020) enforce this property by adapting the parameterized autoregressive Bernoulli probability $q_\theta(X_i|X_{<i}) = \widehat{q_\theta}(X_{<i})^{X_i}(1 - \widehat{q_\theta}(X_{<i}))^{1-X_i}$ by setting $\widehat{q_\theta}(X_{<i}) := \text{clip}(q_\theta(X_{<i}), \epsilon, 1 - \epsilon)$. This adapted probability is then used in NMCMC and NS-NIS to ensure asymptotically unbiased sampling. In Sec. 3.2 we will propose a way how to realize asymptotically unbiased sampling with diffusion models which is experimentally validated in Sec. 5.2.

**Asymptotically Unbiased Sampling with Diffusion Models:** We propose to address asymptotically unbiased sampling with diffusion models by introducing a sampling bias $\epsilon_t$ at each diffusion step $t$. This sampling bias is then used to smooth out the output probability of the conditional diffusion step $q_\theta(X_{t-1}|X_t) = \prod_i^N q_\theta(X_{t-1,i}|X_t)$, where $q_\theta(X_{t-1,i}|X_t) = \widehat{q_\theta}(X_t)_i^{X_{t-1,i}}(1 - \widehat{q_\theta}(X_t)_i)^{(1-X_{t-1,i})}$ and $\widehat{q_\theta}(X_t)_i := \text{clip}(q_\theta(X_t)_i, \epsilon_t, 1 - \epsilon_t)$. By choosing a sampling bias $\epsilon_t > 0$ asymptotically unbiased sampling is ensured.

In practice, we have not found any $\epsilon$ for autoregressive models or diffusion models, which has improved the model in the setting of unbiased estimation.

### A.3.4 MARKOV CHAIN CONVERGENCE CRITERION

To assess the convergence of MCMC chains, we use the integrated autocorrelation time, $\tau_O$, which quantifies the correlation between samples of a chain Sokal (1996). It is defined as:

$$\tau_O = \sum_{\tau=-\infty}^{\infty} \rho_O(\tau),$$

where $\rho_O(\tau)$ is the normalized autocorrelation function of the stochastic process generating the chain for a quantity $f$. For a finite chain of length $N$, the normalized autocorrelation function $\rho_O(\tau)$ is approximated as:

$$\hat{\rho}_O(\tau) = \frac{\hat{c}_O(\tau)}{\hat{c}_O(0)},$$

where

$$\hat{c}_O(\tau) = \frac{1}{L_C - \tau} \sum_{l=1}^{L_C-\tau} (O_l - \mu_O)(O_{l+\tau} - \mu_O), \quad \mu_O = \frac{1}{L_C} \sum_{l=1}^{L_C} O_l$$

and $L_C$ is the length of the Markov chain. Rather than summing the autocorrelation estimator $\hat{\rho}_O(\tau)$ up to $L_C$, which introduces noise as $L_C$ is finite, we truncate the sum at $K \ll L_C$ to balance variance and bias. The integrated autocorrelation time $\hat{\tau}_O$ is then estimated as:

$$\hat{\tau}_O(K) = 1 + 2 \sum_{\tau=1}^{K} \hat{\rho}_O(\tau),$$

where $K$ is chosen as $K \geq C\tau_O$ for a constant $C = 5$, following the recommendations of Sokal (1996).

### A.3.5 FKL W/ MC ALGORITHM

The following pseudocode shows how we minimize the *fKL w/ MC* objective for an unconditional generative diffusion model.

---

**Algorithm 1** Diffusion Model Training based on *fKL w/ MC*

---

1: initialize learning rate $\eta$, number of diffusion trajectories $N$,
2: diffusion trajectory mini-batch size $n$, and mini-batch diffusion time step size $\tau$
3: $\mathcal{D}_B = \emptyset$
4: **for** each epoch in epochs **do**
5:     Sample $X_{0:T}^{0:N} \sim q_\theta(X_{0:T})$
6:     Store $(X_{0:T}^{0:N}, q_\theta(X_{0:T})^{0:N}, \{0, ..., T\}^{0:N})$ in data buffer $\mathcal{D}_B$
7:     **while** data buffer not empty **do**
8:         sample $\{\tau\}^{0:n} := \{t_1, ..., t_\tau\}^{0:n} \sim \mathcal{D}_B$ w/o replacement
9:         obtain corresponding $(X_{0:T}^{0:n}, q_{\theta_{\text{old}}}(X_{0:T})^{0:n})$
10:         compute importance weights $w(X_{0:T}^i) = \frac{\widehat{w}(X_{0:T}^i)}{\sum_j \widehat{w}(X_{0:T}^j)}$ with $\widehat{w}(X_{0:T}^i) = \frac{\widehat{p}(X_{0:T}^i)}{q_{\theta_{\text{old}}}(X_{0:T}^i)}$.
11:         compute loss $L(\theta) = -T \sum_i \sum_{t \in \{\tau\}^i} \left[ w(X_{0:T}^i) \nabla_\theta \log q_\theta(X_{t-1}^i | X_t^i) \right]$
12:         Update $\theta \leftarrow \theta - \eta \nabla_\theta L(\theta)$
13:     **end while**
14: **end for**

---

In UCO we additionally condition the generative model on the CO problem instance and the extension of this algorithm to a conditional generative diffusion model is trivial.

### A.3.6   PPO ALGORITHM

The following pseudocode shows how we minimize the *rKL w/ RL* objective for an unconditional generative diffusion model. In the following, $^{0:N}$ denote indices of different samples so that, for example, $X_{0:T}^{0:N} = (X_{0:T}^1, ..., X_{0:T}^N)$.

In PPO, an additional set of hyperparameters is introduced: $\alpha$, $\lambda$, $c_1$, and $\kappa$. Here, $\alpha$ is the moving average parameter, which is used to compute the rolling average and standard deviation of the reward. $\lambda$ is the trace-decay parameter used to compute eligibility traces in the temporal difference learning algorithm, TD($\lambda$). $c_1$ is the relative weighting between the loss of the value function $L_V(\theta)$ and the loss of the policy $L_\pi(\theta)$, so that the overall loss $L_{\text{PPO}}(\theta)$ is computed with $L_{\text{PPO}}(\theta) = (1 - c_1)L_\pi(\theta) + c_1 L_V(\theta)$.

The value function loss $L_V(\theta)$ is defined as the squared error between the predicted value of the state and the TD($\lambda$)-estimated return:

$$L_V(\theta) = \frac{1}{2}\mathbb{E}_t \left[ (V_\theta(X_t) - G_t^\lambda)^2 \right],$$

where $V_\theta(X_t)$ is the predicted value of state $X_t$, and $G_t^\lambda$ is the TD($\lambda$)-estimated return, which combines the immediate rewards and bootstrapped value estimates of future states.

The TD($\lambda$) return $G_t^\lambda$ is computed as:

$$G_t^\lambda = (1 - \lambda) \sum_{n=1}^{T-t} \lambda^{n-1} G_t^{(n)},$$

where $G_t^{(n)}$ is the $n$-step return defined as:

$$G_t^{(n)} = r_t + \gamma r_{t+1} + \cdots + \gamma^{n-1} r_{t+n-1} + \gamma^n V_\theta(X_{t+n}),$$

with $\gamma$ being the discount factor which we always set to 1.0.

Finally, $\kappa$ is the value that is used for clipping in the policy loss function, which is given by:

$$L_\pi(\theta) = \mathbb{E}_t \left[ \min(r_t(\theta)\hat{A}_t, \text{clip}(r_t(\theta), 1 - \kappa, 1 + \kappa)\hat{A}_t) \right],$$

where $\hat{A}_t$ is the normalized estimator of the advantage function at time step $t$, and $r_t(\theta) = \frac{q_\theta(X_{t-1}|X_t)}{q_{\theta_{\text{old}}}(X_{t-1}|X_t)}$.

The advantage function $A_t$ is computed using TD($\lambda$), and represents how much better an action is compared to the expected value of a given state. It is given by:

$$A_t = G_t^\lambda - V_\theta(X_t),$$

where $G_t^\lambda$ is the TD($\lambda$) return estimate, which is a weighted sum of multi-step returns, and $V_\theta(X_t)$ is the value function. $\hat{A}_t$ is computed by normalizing the advantage for each batch.

---

**Algorithm 2** Diffusion Model Training based on *rKL w/ RL*

---

1: initialize learning rate $\eta$, Number of diffusion trajectories $N$, mini-batch sizes $n, \tau$
2: initialize PPO hyperparameters $\alpha$, $\lambda$, $c_1$, $\kappa$
3: $\mathcal{D}_B = \emptyset$
4: **for** each epoch in epochs **do**
5:      Sample $X_{0:T}^{0:N} \sim q_\theta(X_{0:T})$
6:      Store $(X_{0:T}^{0:N}, q_\theta(X_{0:T})^{0:N}, \{0, ..., T\}^{0:N}, R_{0:T}^{0:N}, V_{0:T}^{\theta,0:N})$ in data buffer $\mathcal{D}_B$
7:      update moving average statistics of the reward using $R_{0:T}^{0:N}$, $\alpha$ and previous statistics
8:      normalize reward according to moving averages
9:      compute estimates of $A_{0:T}^{0:N}$
10:      compute $\hat{A}_{0:T}^{0:N}$ by normalizing $A_{0:T}^{0:N}$
11:      **while** data buffer not empty **do**
12:          sample $\{\tau\}^{0:n} := \{t_1, ..., t_\tau\}^{0:n} \sim \mathcal{D}_B$ w/o replacement
13:          Update $\theta \leftarrow \theta - \eta\nabla_\theta L_{\text{PPO}}(\theta)$
14:      **end while**
15: **end for**

---

## A.4 ARCHITECTURES

### A.4.1 GNN ARCHITECTURE

The architecture we employ is a simple Graph Neural Network (GNN). The process begins with a linear transformation of each node's input features using a layer of $n_h$ neurons. These transformed features, now serving as node embeddings, are then multiplied by a weight matrix also consisting of $n_h$ neurons. Following this, a variance preserving aggregation (Schneckenreiter et al., 2024) is performed over the neighborhood of each node. After message aggregation, we apply the Graph Normalization (Cai et al., 2021). To preserve original node information, a skip connection is incorporated for each node. The aggregated node data, combined with the skip connection, is then processed through a Node Multi-Layer Perceptron. This sequence of operations constitutes a single message passing step, which is repeated $n$ times. After completing all message-passing steps, each resulting node embedding is input into a final three-layer MLP. This MLP computes the probabilities for each solution variable $X_i$. To normalize the data and improve training stability, we apply Layer Normalization Ba et al. (2016) after every MLP layer, with the exception of the final layer in the terminal MLP. When we train the objective using RL methods, we compute the value function by applying an additional three-layer value network on top of a global variance-preserving graph aggregation Schneckenreiter et al. (2024). Here, we use $n_h$ neurons, except in the last layer where only one output neuron is used. Across all our experiments, we consistently use $n_h = 64$ neurons in the hidden layers.

### A.4.2 U-NET ARCHITECTURE

For the experiment on the Ising model, we use a simple U-Net architecture Ronneberger et al. (2015) with overall three convolutional blocks which consist of two convolutional layers with a kernel size of $3 \times 3$ each. After the first convolutional block, we apply a max pooling operation with a window size of $2 \times 2$. The second convolutional block is applied to the downsampled grid. Finally, after upsampling the last convolutional block is applied. For the diffusion model, we then apply a three-layer neural network with $64$ neurons in the first two layers on each node of the grid, which predicts the Bernoulli probability of each state variable. For the autoregressive network, we apply a mean aggregation on all of the nodes which we then put into a three-layer neural network to predict the Bernoulli probability of the next state variable.

## A.5 CO PROBLEM TYPES

All CO problem types considered in this paper are given in Tab. 5.

**Maximum Independent Set:** The Maximum Independent Set problem is the problem of finding the largest set of nodes within a graph under the constraint that neighboring nodes are not in the set.

**Maximum Cut:** The Maximum Cut problem is the problem of partitioning all nodes of a graph into two sets so that the edges between these two sets are as high as possible.

**Minimum Dominating Set:** The Minimum Dominating Set problem is the problem of finding the smallest set of nodes so that every node in the graph is either in the set or adjacent to at least one node in the set.

**Maximum Clique:** The Maximum Clique problem is the problem of finding the largest set of nodes within a graph so that every node within the set is connected to every other node within the set.

| Problem Type | Objective: $\min_{X \in \{0,1\}^N} H(X)$ |
|:---:|:---:|
| MIS | $H(X) = -A \sum_{i=1}^{N} X_i + B \sum_{(i,j) \in \mathcal{E}} X_i \cdot X_j$ |
| MDS | $H(X) = A \sum_{i=1}^{N} X_i + B \sum_{i=1}^{N} (1 - X_i) \prod_{j \in \mathcal{N}(j)} (1 - X_j)$ |
| MaxCl | $H(X) = -A \sum_{i=1}^{N} X_i + B \sum_{(i,j) \notin \mathcal{E}} X_i \cdot X_j$ |
| MaxCut | $H(\sigma) = -\sum_{(i,j) \in \mathcal{E}} \frac{1 - \sigma_i \sigma_j}{2}$   where $\sigma_i = 2X_i - 1$ |

Table 5: Table with energy functions of the MIS, MDS, MaxCl and MaxCut problems (Lucas, 2014). Choosing $A < B$ ensures that all minima of the energy function are feasible solutions. In all of our Experiments, we chose $A = 1.0$ and $B = 1.1$. The table is taken from Sanokowski et al. (2024).

## A.6 GRAPH DATASETS

**RB dataset:** In the RB model, each graph is generated by specifying generation parameters $n, k$, and $p$. With n the number of cliques, i.e. a set of fully connected nodes, and with $k$ the number of nodes within the clique is specified. $p$ serves as a parameter that regulates the interconnectedness between cliques. The lower the value of p the more connections are randomly drawn between cliques. If $p = 1$ there are no connections between the cliques at all. To generate the RB-100 dataset with graphs of an average node size of 100, we generate graphs with $n \in \{9, ..., 15\}$, $k \in \{8, ..., 11\}$, and $p \in [0.25, ..., 1]$. On the RB-small dataset $k \in \{5, ..., 12\}$ and $n \in \{20, ..., 25\}$ and graphs that are smaller than 200 nodes or larger than 300 nodes are resampled. On BA-large $k \in \{20, ..., 25\}$ and $n \in \{40, ..., 55\}$ and graphs that are smaller than 800 nodes or larger than 1200 nodes are resampled. For both of these datasets $p \in [0.3, 1]$.

**Barabasi-Albert dataset:** The BA dataset is generated using the networkx graph library Hagberg & Conway (2020) with the generation parameter $m = 4$. In BA-small number of nodes within each graph is sampled within the range $\{200, ..., 300\}$, and in BA-large number of nodes is sampled within the range of $\{800, ..., 1200\}$.

Ultimately, the matrix $Q$ in $\mathcal{D}(Q)$ can be interpreted as a weighted adjacency matrix. for each CO problem instance, this adjacency matrix is defined by the corresponding graphs of each graph dataset as described in App. A.6, and its weights are given by the CO problem type definition as described in App. A.5.

## A.7 EXPERIMENTS

### A.7.1 UCO EXPERIMENTAL DESIGN

The experiments in Sec. 5.1 are designed to maintain consistent memory requirements, gradient update steps, and training time across all objectives. For DiffUCO, we fix these requirements by setting the batch size to $n_B \times n_G \times n_{\text{diff}}$, where $n_B$ is the number of independently sampled trajectories, $n_G$ is the number of distinct CO problem instances, and $n_{\text{diff}}$ is the number of diffusion steps in

each batch. For the *rKL w/ RL* and *fKL w/ MC* objectives, we can use a minibatch of diffusion steps $n_{\Delta\text{diff}}$, which is not possible with DiffUCO. This allows us to increase the number of diffusion steps by a factor of $k$ while maintaining the same memory requirements during backpropagation. The batch size for these methods becomes $n_B \times n_G \times n_{\Delta\text{diff}}$, where $n_{\Delta\text{diff}} = n_{\text{diff}}/k$. As the number of diffusion steps increases by a factor of $k$, the training time would normally increase accordingly. To counteract this and keep training time consistent across all objectives, we adjust the batch parameters by decreasing $n_B$ by a factor of $k$ and increasing $n_G$ by the same factor. All computations across the batch size are conducted in parallel, and by increasing the batch size the number of updates per epoch is reduced, which reduces the training time per epoch. Therefore, these adjustments maintain a constant overall batch size while reducing training time due to the increased $n_G$. This experimental design ensures that memory requirements, gradient update steps, and training time remain constant across all objectives. It is important to note that with DiffUCO, it is not possible to increase the number of diffusion steps while keeping the training time constant at constant memory requirements. This is because memory requirements would increase by a factor of $k$, which can only be mitigated by reducing either $n_G$ or $n_B$ by a factor of $k$. Reducing $n_G$ would increase the training time by an additional factor of $k$, leading to an overall increase of $k^2$ in training time. Reducing $n_B$ would not decrease the training time as these computations happen in parallel. Consequently, in this case, the computational time overhead would increase by a factor of $k$. Across all of these experiments, the architecture remains the same, except for the experiments with *SDDS: rKL w/ RL*, where an additional variance preserving aggregation Schneckenreiter et al. (2024) on all nodes is applied after the last message passing layer. After that, this embedding is fed into a small Value MLP network with two layers.

In all of the UCO experiments we run on each dataset iterative hyperparameter tuning of the learning rate $\eta$ and starting temperature $\mathcal{T}_{\text{start}}$ for a short annealing duration of $N_{\text{tuning}}$. Here, we first find an optimal $\eta_{\text{opt}}$ such that we find a $\eta_1$ and $\eta_1$ for which $\eta_1 \leq \eta_{\text{opt}} \leq \eta_2$ but the average solution quality is best for $\eta_{\text{opt}}$. After that, we do the same for $\mathcal{T}_{\text{start}}$. We always chose $N_{\text{tuning}} = 500$ except for the RB-large MIS dataset, where we chose $N_{\text{tuning}} = 200$ due to higher computational demands. After obtaining the best hyperparameters we run the final experiments on three different seeds for $N_{\text{Anneal}}$ epochs (see. App.A.10).

### A.7.2 ABLATION ON LEARNING RATE SCHEDULE AND GRAPH NORM LAYER

We provide an ablation on the learning rate schedule (see App. A.10) and the graph normalization layer. The comparison is conducted on MIS on a small dataset of RB-graphs with an average graph size 100 nodes. For each setting, we optimize iteratively first the temperature and then the learning rate for annealing runs with 500 annealing steps as we have done in App. A.7.1. Results are shown in Tab. 6, where we see that the cosine learning rate schedule and the Graph Normalization layer both lead to significantly better models.

| DiffUCO: CE | vanilla | w/ Graph Norm | w/ lr schedule |
|---|---|---|---|
| RB-100 MIS ↑ | $9.63 \pm 0.05$ | $\mathbf{9.71 \pm 0.02}$ | $\mathbf{9.73 \pm 0.04}$ |

Table 6: Ablation on learning rate schedule and Graph Normalization layer on the RB-100 MIS dataset. The larger the MIS size the better. The discrete diffusion model is trained on the dataset without the learning rate schedule (vanilla), with the learning rate schedule (w/ lr schedule), and with the Graph Normalization layer (w/ Graph Norm). Average MIS size is shown over three independent seeds. The standard error is calculated over two independent seeds.

### A.7.3 UNBIASED SAMPLING EXPERIMENTAL DESIGN

For all of our experiments on the Ising model we follow (Nicoli et al., 2020) and use an annealing schedule which is given by $T(n) = \frac{1}{\beta_c}\frac{1}{1-0.998^{h(n+1)}}$, where $n$ is the current epoch and $h$ is a hyperparameter that defines how fast the temperature decays to the target temperature $\frac{1}{\beta_c}$. In our experiments on the Ising model, we keep the overall memory for each method the same. Each experiment fits on an A100 NVIDIA GPU with 40 GB of memory. In unbiased sampling, we use 400 iterations for NMCMC with a batch size of 1200 states. In SN-NIS we estimate the observables

with 480.000 states. Nicoli et al. (2020) use 500.000 states in their experiments. Error bars are calculated over three independent SN-NIS and MCMC runs.

## A.8 Additional Experiments

### A.8.1 Study on Number of Diffusion Steps and Memory Requirements

We provide further experiments on the RB-small MIS problem, evaluating the relative error $\epsilon_{\text{rel}} := \frac{|E_{\text{opt}} - E_{\text{model}}|}{|E_{\text{opt}}|}$ and the best relative error $\epsilon_{\text{rel}}^* := \frac{|E_{\text{opt}} - E_{\text{model}}^*|}{|E_{\text{opt}}|}$, where $E_{\text{opt}}$ is the optimal set size on this dataset and $E_{\text{model}}$ is the average and $E_{\text{model}}^*$ the best-set size out of 60 states of the trained model. We train DiffUCO, SDDS: rKL w/ RL, and SDDS: fKL w/ MC for 2000 epochs and plot these metrics over an increasing number of diffusion steps. The results are shown in Fig. 1. We train each method on 4, 8, 12, and 16 diffusion steps, keeping the overall batch size constant for each method. For DiffUCO, the memory requirements scale linearly with the number of diffusion steps, as indicated by the size of the marker in Fig. 1. In contrast, for SDDS: rKL w/ RL and SDDS: fKL w/ MC, we keep the mini-batch size fixed at 4, so the memory requirement does not increase, hence the marker size stays the same. Specifically, the memory requirements are here the same as for DiffUCO with 4 diffusion steps. We observe that for DiffUCO and SDDS: rKL w/ RL of the methods $\epsilon_{\text{rel}}$ and $\epsilon_{\text{rel}}^*$ improved with an increasing number of diffusion steps and that SDDS: rKL w/ RL performs better than DiffUCO in most cases. For SDDS: fKL w/ MC $\epsilon_{\text{rel}}$ does not improve after 12 diffusion steps. We additionally show in Tab. 7 the runtime per epoch for each run in Fig. 1 which shows empirically that SDDS: rKL w/ RL enables superior trade-offs between training time and memory requirements. For instance, SDDS: rKL w/ RL with 12 diffusion steps exhibits a better performance than DiffUCO with 16 diffusion steps while consuming slightly less training time (see Tab. 7) and four times less memory (see Fig. 1).

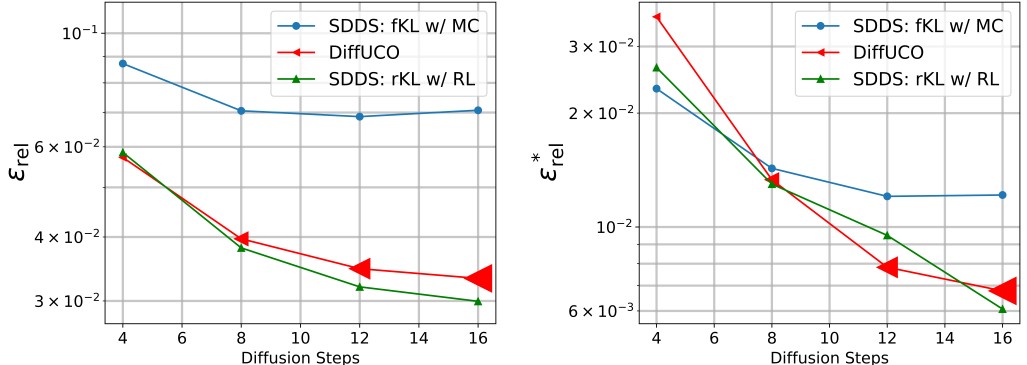

Figure 1: $\epsilon_{\text{rel}}$ (left) and $\epsilon_{\text{rel}}^*$ (right) on the MIS RB-small dataset over an increasing amount of diffusion steps. The marker size is proportional to the memory requirements that are needed during training.

| Method | Diffusion Steps | 4 | 8 | 12 | 16 |
|---|---|---|---|---|---|
| DiffUCO | Runtime (d:h:m) | 0:12:46 | 0:22:13 | 1:09:53 | 1:21:00 |
| SDDS: rKL w/ RL | Runtime (d:h:m) | 0:16:06 | 1:07:06 | 1:20:26 | 2:11:26 |
| SDDS: fKL w/ MC | Runtime (d:h:m) | 0:16:06 | 1:04:06 | 1:22:40 | 2:08:06 |

Table 7: Comparison of training time in d:h:m for different methods across various diffusion steps on the experiment from Fig. 1.

### A.8.2 Spin Glass Experiments

**Unbiased Sampling:**

We follow Del Bono et al. (2024) and conduct experiments on the Edwards-Anderson (EA) spin glass model in the context of unbiased sampling. Here, this model is defined on a periodic 2-D

grid, where neighboring spins interact with each other via random couplings $J_{ij}$ sampled from a normal distribution with zero mean and variance of one. We consider this problem at $\beta \approx 1.51$ as sampling from this model at this temperature is known to be particularly hard for local MCMC samplers (Ciarella et al., 2023). Since this model cannot be solved analytically, we cannot compare ground truth values for free energy, internal energy, or entropy. Therefore, we use the free energy and the effective sample size as a baseline, as a lower free energy and a larger effective sample size is generally better. We evaluate the performance of DiffUCO, SDDS: rKL w/ RL and SDDS: fKL w/ MC under the same computational constraints. All models use a GNN architecture with 6 message passing steps (see App. A.4) to incorporate the neighboring couplings as edge features. We train each method under the same computational budget and similar training time, which means that we train SDDS: rKL w/ RL and SDDS: fKL w/ MC with 150 diffusion steps for 400 epochs and DiffUCO with 50 diffusion steps and 1200 epochs. In each case, we use 200 samples during training and evaluate the free energy and effective sample size using 480000 samples. The results of these experiments are shown in Tab. 8, where we observe that SDDS: fKL w/ MC performs best in terms of free energy and SDDS: rKL w/ RL performs best in terms of effective sample size.

| EA $16 \times 16$ | Free Energy $\mathcal{F}/L^2 \downarrow$ | $\epsilon_{\text{eff}}/M \uparrow$ |
|---|---|---|
| DiffUCO | $-0.329 \pm 0.008$ | $5.22 \times 10^{-6} \pm 1.3 \times 10^{-6}$ |
| SDDS: rKL w/ RL | $-1.09 \pm 0.003$ | $\mathbf{8.56 \times 10^{-6} \pm 2.29 \times 10^{-6}}$ |
| SDDS: fKL w/ MC | $\mathbf{-1.165 \pm 0.003}$ | $3.2 \times 10^{-6} \pm 4 \times 10^{-7}$ |

Table 8: Free Energy per size and effective sample size per sample of different diffusion samplers on the Edwards-Anderson model of size $16 \times 16$.

**Ground State Prediction:** We also conduct experiments on the Edwards-Anderson model to predict the lowest energy configurations. Here, we follow the setting from Hibat-Allah et al. (2021) and sample neighboring couplings from a uniform distribution $[-1, 1[$ on a 2-D grid of size $10 \times 10$. We train SDDS: rKL w/ RL and SDDS: fKL w/ MC at 100 diffusion steps and 25 mini-batch diffusion steps. We follow Hibat-Allah et al. (2021) and train the model using 25 states and use 10000 equilibrium steps at $T_{\text{start}} = 1.0$ and anneal the temperature down to zero. Our models use a GNN architecture with 8 message passing steps (see App. A.4) and are trained for 4000 training steps. We compare to the result of classical-quantum optimization (CQO) (Martoňák et al., 2002; Gomes et al., 2019; Sinchenko & Bazhanov, 2019; Zhao et al., 2020), Variational Quantum Annealing (VQA), regularized Variational Quantum Annealing (RVQA) and Variational Neural Annealing (VNA) as reported in Hibat-Allah et al. (2021) at the same amount of training steps. Results of the average energy value over 200 samples are shown in Tab. 9, where we see that SDDS: rKL w/ RL significantly outperforms all other methods.

| EA $10 \times 10$ | CQO (r) | VQA (r) | RVQA (r) | VNA (r) | SDDS: rKL w/ RL | SDDS: fKL w/ MC |
|---|---|---|---|---|---|---|
| $\epsilon_{\text{rel}}/L^2 \downarrow$ | $2 \times 10^{-2} \pm 1 \times 10^{-2}$ | $2 \times 10^{-3} \pm 1 \times 10^{-3}$ | $1 \times 10^{-3} \pm 1 \times 10^{-3}$ | $2 \times 10^{-4} \pm 1 \times 10^{-4}$ | $\mathbf{1.98 \times 10^{-5} \pm 4.35 \times 10^{-5}}$ | $8.23 \times 10^{-4} \pm 2.47 \times 10^{-4}$ |

Table 9: Average ground state energies of different diffusion samplers on the 2-D Edwards-Anderson model of size $10 \times 10$. (r) indicates that results are taken from (Hibat-Allah et al., 2021).

### A.8.3 TIME MEASUREMENT

We follow Sanokowski et al. (2024) and perform all time measurements for Deep Learning-based methods on an A100 NVIDIA GPU and perform the time measurement after the functions are compiled with jax.jit.

### A.9 MEMORY REQUIREMENTS

The experimental setups for various datasets and models have specific GPU requirements. For the RB-small dataset, two A100 NVIDIA GPUs with 40GB of memory each are necessary. The RB-large MIS experiment demands four such GPUs. In contrast, the BA-small dataset can be processed using a single A100 GPU, while the BA-large dataset requires two A100 GPUs. The Ising model experiments can be conducted efficiently with one A100 GPU.

## A.10 Hyperparameters

For all of our experiments, we use one iteration of cosine learning rate (Loshchilov & Hutter, 2017) with warm restarts, where we start at a low learning rate of $10^{-10}$ and increase it linearly to a learning rate of $\lambda_{max}$ for 2.5% of epochs. After that, the learning rate is reduced via a cosine schedule to $\lambda_{max}/10$. We use Radam as an optimizer Liu et al. (2020). All hyperparameters and commands to run all of our experiments can be found in the .txt files within our code in /argparse/experiments/UCO and /argparse/experiments/Ising. In our experiments, we always use 6 diffusion steps for DiffUCO and 12 diffusion steps for *SDDS: rKL w/ RL* and *SDDS: fKL w/ MC*, except on the BA-small MDS and BA-small MaxCut dataset where we use 7 and 14 diffusion steps respectively. Compared to (Sanokowski et al., 2024) we use up to a factor of 4 times more diffusion steps, as they use only between 3 and 6 diffusion steps under similar computational constraints. We always keep PPO-related hyperparameters to the default value, except on RB-large MIS, where we have adjusted the hyperparameter $\alpha$ (see App. A.3.6).

## A.11 Code

The code is written in jax (Bradbury et al., 2018).

## A.12 Extended Tables

For completeness we include in Tab. 11, Tab. 10 and in Tab. 12 other baseline methods as reported in Sanokowski et al. (2024).

| **MIS** | | RB-small | | RB-large | |
|---|---|---|---|---|---|
| Method | Type | Size ↑ | time ↓ | Size ↑ | time ↓ |
| Gurobi Gurobi Optimization, LLC (2023) | OR | $20.13 \pm 0.03$ | 6:29 | $42.51 \pm 0.06^*$ | 14:19:23 |
| LwtD (r) (Ahn et al., 2020) | UL | 19.01 | 2:34 | 32.32 | 15:06 |
| INTEL (r) (Li et al., 2018) | SL | 18.47 | 26:08 | 34.47 | 40:34 |
| DGL (r) (Böther et al., 2022b) | SL | 17.36 | 25:34 | 34.50 | 47:28 |
| LTFT (r) (Zhang et al., 2023) | UL | 19.18 | 1:04 | 37.48 | 8:44 |
| DiffUCO (r) (Sanokowski et al., 2024) | UL | $18.88 \pm 0.06$ | 0:14 | $38.10 \pm 0.13$ | 0:20 |
| DiffUCO: CE (r) (Sanokowski et al., 2024) | UL | $19.24 \pm 0.05$ | 1:48 | $38.87 \pm 0.13$ | 9:54 |
| DiffUCO | UL | $19.42 \pm 0.03$ | 0:02 | $39.44 \pm 0.12$ | 0:03 |
| SDDS: *rKL w/ RL* | UL | $\mathbf{19.62 \pm 0.01}$ | 0:02 | $\mathbf{39.97 \pm 0.08}$ | 0:03 |
| SDDS: *fKL w/ MC* | UL | $19.27 \pm 0.03$ | 0:02 | $38.44 \pm 0.06$ | 0:03 |
| DiffUCO: CE | UL | $19.42 \pm 0.03$ | 0:20 | $39.49 \pm 0.09$ | 6:38 |
| SDDS: *rKL w/ RL*-CE | UL | $\mathbf{19.62 \pm 0.01}$ | 0:20 | $\mathbf{39.99 \pm 0.08}$ | 6:35 |
| SDDS: *fKL w/ MC*-CE | UL | $19.27 \pm 0.03$ | 0:19 | $38.61 \pm 0.03$ | 6:31 |

Table 10: Extended result table. Average independent set size on the whole test dataset on the RB-small and RB-large datasets. The higher the better. The total evaluation time is shown in h:m:s. (r) indicates that results are reported as in Sanokowski et al. (2024). $\pm$ represents the standard error over three independent training seeds. (CE) indicates that results are reported after applying conditional expectation. The best neural method is marked as bold. Gurobi results with $^*$ indicate that Gurobi was run with a time limit. On MIS RB-large the time-limit is set to 120 seconds per graph. In this table, SL is for supervised learning and UL is for unsupervised learning methods.

| MDS | | BA-small | | BA-large | |
|---|---|---|---|---|---|
| Method | Type | Size ↓ | time ↓ | Size ↓ | time ↓ |
| Gurobi Gurobi Optimization, LLC (2023) | OR | $27.84 \pm 0.00$ | 1:22 | $104.01 \pm 0.27$ | 3:35:15 |
| Greedy (r) | H | 37.39 | 4:26 | 140.52 | 1:10:02 |
| MFA (r) (Bilbro et al., 1988) | H | 36.36 | 5:52 | 126.56 | 1:13:02 |
| EGN: CE (r) (Karalias & Loukas, 2020) | UL | 30.68 | 2:00 | 116.76 | 7:52 |
| EGN-Anneal: CE (r) (Sun et al., 2022) | UL | 29.24 | 2:02 | 111.50 | 7:50 |
| LTFT (r) (Zhang et al., 2023) | UL | 28.61 | 4:16 | 110.28 | 1:04:24 |
| DiffUCO (r) (Sanokowski et al., 2024) | UL | $28.30 \pm 0.10$ | 0:10 | $107.01 \pm 0.33$ | 0:10 |
| DiffUCO: CE (r) (Sanokowski et al., 2024) | UL | $28.20 \pm 0.09$ | 1:48 | $106.61 \pm 0.30$ | 6:56 |
| DiffUCO | UL | $28.10 \pm 0.01$ | 0:01 | $\mathbf{105.21 \pm 0.21}$ | 0:01 |
| SDDS: *rKL w/ RL* | UL | $\mathbf{28.03 \pm 0.00}$ | 0:02 | $\mathbf{105.16 \pm 0.21}$ | 0:02 |
| SDDS: *fKL w/ MC* | UL | $28.34 \pm 0.02$ | 0:01 | $105.70 \pm 0.25$ | 0:02 |
| DiffUCO: CE | UL | $28.09 \pm 0.01$ | 0:16 | $\mathbf{105.21 \pm 0.21}$ | 1:45 |
| SDDS: *rKL w/ RL*-CE | UL | $\mathbf{28.02 \pm 0.01}$ | 0:16 | $\mathbf{105.15 \pm 0.20}$ | 1:41 |
| SDDS: *fKL w/ MC*-CE | UL | $28.33 \pm 0.02$ | 0:16 | $105.7 \pm 0.25$ | 1:41 |

Table 11: Extended result table. Average dominating set size on the whole test dataset on the BA-small and BA-large datasets. The lower the set size the better. Total evaluation time is shown in h:m:s. (r) indicates that results are reported as in Sanokowski et al. (2024). $\pm$ represents the standard error over three independent training seeds. (CE) indicates that results are reported after applying conditional expectation. The best neural method is marked as bold. In this table, H stands for heuristic, SL for supervised learning and UL for unsupervised learning methods.

| MaxCl | | RB-small | | MaxCut | | BA-small | | BA-large | |
|---|---|---|---|---|---|---|---|---|---|
| Method | Type | Size ↑ | time ↓ | Method | Type | Size ↑ | time ↓ | Size ↑ | time ↓ |
| Gurobi Gurobi Optimization, LLC (2023) | OR | $19.06 \pm 0.03$ | 11:00 | Gurobi (r) | OR | $730.87 \pm 2.35^*$ | 17:00:00 | $2944.38 \pm 0.86^*$ | 2:35:10:00 |
| Greedy (r) | H | 13.53 | 0:50 | Greedy (r) | H | 688.31 | 0:26 | 2786.00 | 6:14 |
| MFA (r) (Bilbro et al., 1988) | H | 14.82 | 1:28 | MFA (r) | H | 704.03 | 3:12 | 2833.86 | 14:32 |
| EGN: CE (r) (Karalias & Loukas, 2020) | UL | 12.02 | 1:22 | EGN: CE (r) | UL | 693.45 | 1:32 | 2870.34 | 5:38 |
| EGN-Anneal: CE (r) (Sun et al., 2022) | UL | 14.10 | 4:32 | EGN-Anneal : CE (r) | UL | 696.73 | 1:30 | 2863.23 | 5:36 |
| LTFT (r) (Zhang et al., 2023) | UL | 16.24 | 1:24 | LTFT (r) | UL | 704 | 5:54 | 2864 | 42:40 |
| DiffUCO (r) (Sanokowski et al., 2024) | UL | $14.51 \pm 0.39$ | 0:08 | DiffUCO (r) | UL | $727.11 \pm 2.31$ | 0:08 | $2947.27 \pm 1.50$ | 0:08 |
| DiffUCO: CE (r) (Sanokowski et al., 2024) | UL | $16.22 \pm 0.09$ | 2:00 | DiffUCO: CE (r) | UL | $727.32 \pm 2.33$ | 2:00 | $2947.53 \pm 1.48$ | 7:34 |
| DiffUCO | UL | $17.40 \pm 0.02$ | 0:02 | DiffUCO | UL | $\mathbf{731.30 \pm 0.75}$ | 0:02 | $\mathbf{2974.60 \pm 7.73}$ | 0:02 |
| SDDS: *rKL w/ RL* | UL | $\mathbf{18.89 \pm 0.04}$ | 0:02 | SDDS: *rKL w/ RL* | UL | $\mathbf{731.93 \pm 0.74}$ | 0:02 | $2971.62 \pm 8.15$ | 0:02 |
| SDDS: *fKL w/ MC* | UL | $18.40 \pm 0.02$ | 0:02 | SDDS: *fKL w/ MC* | UL | $\mathbf{731.48 \pm 0.69}$ | 0:02 | $\mathbf{2973.80 \pm 7.57}$ | 0:02 |
| DiffUCO: CE | UL | $17.40 \pm 0.02$ | 0:38 | DiffUCO: CE | UL | $\mathbf{731.30 \pm 0.75}$ | 0:15 | $\mathbf{2974.64 \pm 7.74}$ | 1:13 |
| SDDS: *rKL w/ RL*-CE | UL | $\mathbf{18.90 \pm 0.04}$ | 0:38 | SDDS: *rKL w/ RL*-CE | UL | $\mathbf{731.93 \pm 0.74}$ | 0:14 | $2971.62 \pm 8.15$ | 1:08 |
| SDDS: *fKL w/ MC*-CE | UL | $18.41 \pm 0.02$ | 0:38 | SDDS: *fKL w/ MC*-CE | UL | $\mathbf{731.48 \pm 0.69}$ | 0:14 | $\mathbf{2973.80 \pm 7.57}$ | 1:08 |

Table 12: Extended result table: Left: Testset average clique size on the whole on the RB-small dataset. The larger the set size the better. Right: Average test set cut size on the BA-small and BA-large datasets. The larger the better. Left and Right: Total evaluation time is shown in d:h:m:s. (r) indicates that results are reported as in Sanokowski et al. (2024). (CE) indicates that results are reported after applying conditional expectation. Gurobi results with $^*$ indicate that Gurobi was run with a time limit. On MDS BA-small the time limit is set to 60 and on MDS BA-large to 300 seconds per graph. The best neural method is marked as bold. In these tables, H stands for heuristic, SL for supervised learning and UL for unsupervised learning methods.

