# OpenReview forum: "Scalable Discrete Diffusion Samplers: Combinatorial Optimization and Statistical Physics"
_ICLR.cc/2025/Conference — ICLR 2025 Poster_

### Official Review · Reviewer_UBbm · 2024-10-30

**Soundness:** 3
**Presentation:** 2
**Contribution:** 3
**Rating:** 6
**Confidence:** 2

**Summary:**

The article considers the task of learning to sample from complex unnormalized distribution for discrete variables, with application to optimization and sampling problems. The problem in this kind of task is the need to keep track of the entire diffusion trajectory, making it very demanding in memory. The authors alleviate this problem by using Importance Sampling (in one case) or Reinforcement Learning techniques in another one. More precisely, they propose two kinds of modification: first to use the forward KL (instead of the reverse one) in order to remove the problem of having to save the whole trajectory, at the cost of needed to use sampling from the target distribution (here done with importance sampling); second they reformulate the optimization problem on the reverse KL as a Reinforcement Learning problem, allowing to deal with the optimization using technique coming from RL.

**Strengths:**

The paper provides two different ways of solving the memory issue in the context of discrete diffusion model when using the reverse KL.
A reasonable list of optimization's benchmarks are provided.

**Weaknesses:**

The benchmark of sampling on Ising is ok, but we could have expected more difficult cases. For instance, the 2D spin glass ($J= \pm 1$) might be
* a more challenging task not too different from the Ising case considered in the paper
* and for which the polynomial algorithm exists at least for the ground states.

In relation to both their work and this case, they can also refer to the following article: https://arxiv.org/pdf/2407.19483 which study using a different approach the sampling on the 2D Ising spin glass for comparison.

**Questions:**

- I guess the reason why the whole trajectory is needed to optimize eq. 2 is because the gradient will be applied to the expectation value, from which all steps will be affected. Is that correct ? if yes, it might be clearer to specify it right after eq. 2 (using main text, footnote or a reference to the supplementary materials).

- In 2.2, the distribution $\mathcal{D}(Q)$ is not specified. It might be useful to put a bit of precision, since I imagine that the goal is to use an ensemble of similar optimization problem. For instance, a reference to sec. 4 of the appendix with a detail on the distribution of Q for the different considered cases.

- What is exactly the difference between IS and SN-NIS ? it would be good for clarity to add an explanation in the article on that point.

- Same comment for neural MCMC, how is it any different from classical MCMC ? it would be good for clarity to add an explanation in the article on that point.

- The part 2.3 is a bit misleading, it seems to describe classical method of sampling while it is not even clear at that point where it will be used for. In fact, most of it is repeated in sec. 3.2...

Minor comment:
- eq. line 114, p should be p_B no ?

---

> ### Author Response · Authors · 2024-11-22
> **Response to Review 1/2**
>
> We thank the reviewer for the thoughtful and constructive review. We address the weaknesses and questions in the following:
>
> **W1: The benchmark of sampling on Ising is ok, but we could have expected more difficult cases. For instance, the 2D spin glass (J=±1) might be**
>
> **-   a more challenging task not too different from the Ising case considered in the paper**
>
> **-   and for which the polynomial algorithm exists at least for the ground states.**
>
>
> At the critical temperature the ferromagnetic (J=1) Ising model is known to exhibit large-range correlations that are notoriously hard to sample. For this reason it is a popular choice in the corresponding literature (Wu et al. (2019), McNaughton et al., (2020), Nicoli et al. (2020), Wu et al. (2021)). Nevertheless, we agree with the reviewer that experiments on spin glasses would be a well motivated and feasible extension of our experimental evaluation. Therefore, we now also include unbiased sampling results (see A.8.2) on the Edwards-Andeson model as studied in Del Bono et al. (2024). We compare our SDDS methods to DiffUCO in terms of free energy and effective sample size. However, we cannot compare our results to Del Bono et al. (2024) since their evaluation metrics require the availability of a data set that has to be generated via extensive MCMC runs which we cannot reproduce in the limited time period of the rebuttal. In addition, such a comparison would not be very insightful since their method is trained using this data set and hence their method is tackling a fundamentally different problem as ours.
> We now also include a UCO experiment on an Edwards-Andeson spin glass model. Here we follow Hibat-Allah et al (2021) and benchmark our methods on their ground state search problem for this system (A.8.2, Tab. 9). These experiments show that SDDS: w/ RL and SDS fKL w/ MC are highly performant methods on this problem.
>
>
>
> **Q: “I guess the reason why the whole trajectory is needed to optimize eq. 2 is because the gradient will be applied to the expectation value, from which all steps will be affected. Is that correct ? if yes, it might be clearer to specify it right after eq. 2 (using main text, footnote or a reference to the supplementary materials).”**
> Yes, this is correct. As proposed by the reviewer we have added a clarification in  L.154f.
>
>
>
>
> **Q: “In 2.2, the distribution D(Q) is not specified. It might be useful to put a bit of precision, since I imagine that the goal is to use an ensemble of similar optimization problem. For instance, a reference to sec. 4 of the appendix with a detail on the distribution of Q for the different considered cases.”**
> We thank the reviewer for pointing this out. We have added a definition of D(Q) in L.168ff. and added a reference to the experiment section and the appendix. In A.6 we have added more details about the distribution and the corresponding graph datasets and how the CO problem type in combination with the graph adjacency relates to the distribution D(Q).
>
>
>
>
> **Q: “What is exactly the difference between IS and SN-NIS ? it would be good for clarity to add an explanation in the article on that point.”
> “Same comment for neural MCMC, how is it any different from classical MCMC ? it would be good for clarity to add an explanation in the article on that point.”**
> We follow the suggestion of the reviewer and improve the presentation of these methods. We add corresponding sections in the appendix that introduce the IS, NIS, SN-NIS (A.1.1, A.2), and details about our application of SN-NIS with diffusions models were already present at the point of submission in A.2.1. Similarly, we introduce now MCMC, NMCMC in more detail in A.2.2 and A.2.3, and our application of NMCMC to diffusion models was already present in A.2.4.

---

> ### Author Response · Authors · 2024-11-22
> **Response to Review 2/2**
>
> **Q: “The part 2.3 is a bit misleading, it seems to describe classical method of sampling while it is not even clear at that point where it will be used for. In fact, most of it is repeated in sec. 3.2...”**
>
> The significance of Sec. 2.3 is to introduce the reader to the problem of unbiased sampling and the ML-based methods SN-NIS and NMCMC that have been proposed in the literature to tackle this problem. Describing these methods is crucial in the context of our work since these two methods are the basis for our newly introduced diffusion-based unbiased sampling methods which are introduced in Sec. 3.2. In the reviewed version of this paper we state explicitly at the end of Sec. 2.3 that SN-NIS and NMCMC themselves are not applicable with approximate likelihood models like diffusion models. Hence we would like to stress that Sec. 2.3 and Sec. 3.2 are by no means repetitive since they introduce different methods - the ones presented in Sec. 3.2 being in fact a novel contribution of our work.
> However, due to this critique of the reviewer we recognize that the significance of Sec. 2.3 and Sec. 3.2 and their connection should be presented more prominently than in the end of Sec 2.3. Hence we added a clarifying explanation in the first paragraph of both, Sec. 2.3 and Sec. 3.2.
>
>
>
> **Q: Minor Comment: “eq. line 114, p should be p_B no ?”**
> We thank the reviewer for pointing this out and have corrected this in the updated manuscript.
>
> In conclusion, we thank the reviewer for highlighting several shortcomings. We are convinced that we could use this feedback to improve our work significantly and hope that these improvements will be reflected in corresponding updates of the reviewer's rating.
>
>
>
> **References:**
>
> Del Bono, L. M., Ricci-Tersenghi, F., & Zamponi, F. (2024). Nearest-Neighbours Neural Network architecture for efficient sampling of statistical physics models. arXiv preprint arXiv:2407.19483.
>
>
>
> Wu, Dian, Lei Wang, and Pan Zhang. "Solving statistical mechanics using variational autoregressive networks." Physical review letters 122.8 (2019): 080602.
>
>
>
> McNaughton, B., et al. "Boosting Monte Carlo simulations of spin glasses using autoregressive neural networks." Physical Review E 101.5 (2020): 053312.
>
>
>
> Nicoli, Kim A., et al. "Asymptotically unbiased estimation of physical observables with neural samplers." Physical Review E 101.2 (2020): 023304.
>
> Wu, Dian, Riccardo Rossi, and Giuseppe Carleo. "Unbiased Monte Carlo cluster updates with autoregressive neural networks." Physical Review Research 3.4 (2021): L042024.
>
> Hibat-Allah, M., Inack, E. M., Wiersema, R., Melko, R. G., & Carrasquilla, J. (2021). Variational neural annealing. Nature Machine Intelligence, 3(11), 952-961.

---

> > ### Comment · Reviewer_UBbm · 2024-11-25
> >
> > I thank the authors for their detailed answer. Should that be possible I would raise my score to 7.

---

> > > ### Author Response · Authors · 2024-11-26
> > > **Answer to comment**
> > >
> > > We appreciate your careful review and positive feedback on our paper. We are pleased that our rebuttal has successfully addressed your initial concerns. We noticed your intention to raise the score to 7, which is unfortunately not possible within the current review scoring system.
> > >
> > > Given that our responses have addressed your previous comments, we would welcome any additional specific feedback. If you have no further substantive criticisms about our methodology or experimental design, we kindly request the reviewer to consider raising the score to 8. This would more accurately reflect the contributions of our work and the improvements that we have provided during the rebuttal phase.

---

### Official Review · Reviewer_RMkz · 2024-11-04

**Soundness:** 3
**Presentation:** 2
**Contribution:** 2
**Rating:** 6
**Confidence:** 3

**Summary:**

This paper introduces Scalable Discrete Diffusion Samplers (SDDS) for generating samples from complex, unnormalized distributions over discrete domains, with applications in combinatorial optimization and statistical physics. Traditional methods in this area are limited by memory scaling issues and lack of unbiased sampling capability. To address these challenges, the authors propose two new training methods for diffusion samplers: one based on policy gradient reinforcement learning (RL) for minimizing the reverse KL divergence and another using Self-Normalized Neural Importance Sampling (SN-NIS) for unbiased forward KL divergence estimation. These approaches enable more diffusion steps within fixed memory constraints and allow for unbiased sampling, which is crucial for many scientific applications. Experiments on combinatorial optimization benchmarks show that SDDS achieves state-of-the-art performance, and unbiased sampling evaluations on the Ising model demonstrate improved accuracy over traditional autoregressive models.

**Strengths:**

- The paper introduces two novel methods (RL-based reverse KL and SN-NIS-based forward KL) to address memory scaling issues, enabling the use of more diffusion steps while staying within fixed memory constraints.
- By extending importance sampling and Markov Chain Monte Carlo methods to diffusion models, the authors enable unbiased sampling—a critical requirement for scientific applications that require accurate expectation estimates.

**Weaknesses:**

- Although the paper reports inference time alongside results, a more comprehensive analysis of computational costs is needed, particularly in terms of training time and overall efficiency.
- The paper would benefit from a deeper theoretical exploration of the convergence properties of the proposed methods, especially the RL-based approach. Clarifying the conditions under which convergence is guaranteed would strengthen the paper's contributions.
- The experiments lack ablation studies to isolate the effects of individual components of their models.
- The paper primarily compares the proposed methods with DiffUCO.
- The presentation of the experimental results could benefit from clearer organization and more detailed explanations to improve readability and interpretability.

**Questions:**

- How do the proposed methods affect overall computational efficiency? Specifically, does the integration of PPO in SDDS: rKL w/ RL significantly increase training time compared to standard diffusion models or autoregressive approaches?
- Why was PPO specifically chosen for optimizing the rKL-based objective?
- Can the authors provide theoretical or empirical insights into the convergence properties of the proposed methods? Specifically, under what conditions do the RL-based training procedures converge, and how stable are they across different problem instances?
- How does varying the number of diffusion steps during training and inference influence the model's performance? Is there an optimal range of steps beyond which improvements plateau or diminish?
- Why was DiffUCO the primary(or only) comparison, and have the authors considered additional benchmarks such as those in Zhang et al. (2023)? Given that MDS results might not be state-of-the-art, how would the method perform against others besides DiffUCO?

[1] Dinghuai Zhang, Hanjun Dai, Nikolay Malkin, Aaron C. Courville, Yoshua Bengio, and Ling Pan. Let the flows tell: Solving graph combinatorial problems with gflownets. NeurIPS 2023

---

> ### Author Response · Authors · 2024-11-22
> **Response to review 1/3**
>
> We thank the reviewer for the thoughtful and constructive review. We address the weaknesses and questions in the following:
>
>
>
> **W1: “Although the paper reports inference time alongside results, a more comprehensive analysis of computational costs is needed, particularly in terms of training time and overall efficiency.”
> and
> W3: “The experiments lack ablation studies to isolate the effects of individual components of their models.”**
>
> We emphasize that we present ablation studies for every single experiment we do. We always ablate the usage of our methodological contributions:
>
>
> -   PPO for rKL (rKL w/ RL vs DiffUCO),
>
> -   fKL (vs rKL),
>
> -   Conditional Expectation (CE, not a novel contribution but shown to enhance performance).
>
>
>
>
> In addition, we now include a dedicated study of the performance of SDDS methods and DiffUCO in terms of varying numbers of diffusion steps and memory usage in Fig. 1. In Tab. 7 we additionally report the corresponding training times along with a discussion in the text of A.8.1.
>
> These extensive ablations and comparisons unambiguously support the methodological contributions of our work. In case the reviewer still finds a lack in our experimental validation, we would kindly ask the reviewer to explicitly name missing ablations in order to give us the chance to improve our experimental validation.
>
>
>
> **W2: “The paper would benefit from a deeper theoretical exploration of the convergence properties of the proposed methods, especially the RL-based approach. Clarifying the conditions under which convergence is guaranteed would strengthen the paper's contributions.”**
> While this would indeed be an interesting aspect we consider it out of scope for this work and emphasize that none of the related works in this field theoretically investigate this question. This question is not specific to NPO but about the general convergence of RL-methods, actor-critic methods in particular. Recent work (Tian et al., 2023) provides convergence rates for actor-critic methods (like PPO) under several idealized assumptions on e.g. the network architecture and timescales of the learning dynamics of the actor and the critic. Importantly, these analyses are not directly applicable to our setting since their assumption on e.g. these timescales and on the boundedness of the rewards cannot be trivially guaranteed throughout training (bounding our reward in L. 269 is not possible without imposing further assumptions). Due to these challenges convergence properties are addressed in dedicated works and do not fit into the scope of ours.
>
>
>
>
>
> **W4: “The paper primarily compares the proposed methods with DiffUCO.”**
> As explained above the comparison to DiffUCO is crucial since it represents not only a comparison to the SOTA method but also an ablation on one of our methodological contributions (improved performance and memory efficiency through the application of PPO in SDDS: rKL w/ RL). If the reviewer means with “primarily” that our comparison to other methods is insufficient, we would like to point out that all experiments that compare SDDS to DiffUCO also include comparisons against LTFT (Zhang et al., 2023) and Gurobi. DiffUCO appears only more often in our result tables because we do not only report the literature results but we re-ran the experiments for this method ourselves and also report our results with identical computational resources. In addition, we do these comparisons for the versions of SDDS and DiffUCO with and without Conditional Expectation (CE).
>
> We do not include other baselines from Sanokowski et al. 2024 in the tables because of the page limitation of the main text which necessitated that we omit some results of prior works which are not SOTA. For completeness we now provide extended result tables of our UCO experiments in the A.12 (Tab. 10, Tab. 11 and Tab.12). We now also include the results of an additional sampling method (VAN, Wu et al. (2019)) in Tab. 4. In addition, as requested by Reviewer UBbm we now also present results for a ground state search on an Edward-Anderson spin-glass model (see A.8.2). Here we compare our method to the work of Hibat-Allah et al. (2021).
>
>
> **W5: “The presentation of the experimental results could benefit from clearer organization and more detailed explanations to improve readability and interpretability.”**
> We made several improvements to the text in the experimental section. We would be very happy to improve our presentation if the reviewer could be more specific about the text parts which could be improved in terms of readability and interpretability.

---

> ### Author Response · Authors · 2024-11-22
> **Response to review 2/3**
>
> **Q1: “How do the proposed methods affect overall computational efficiency? Specifically, does the integration of PPO in SDDS: rKL w/ RL significantly increase training time compared to standard diffusion models or autoregressive approaches?”**
> The integration of PPO in SDDS: rKL w/o RL introduces the possibility of reducing the memory requirements without changing the batch size model size or the number of diffusion steps T.
>
> A method like DiffUCO needs to keep the computational graphs for all diffusion steps in the memory in order to calculate gradients. This is in contrast to SDDS: rKL w/o RL which builds up a buffer of trajectories and then samples only a fixed number of diffusion steps at once to calculate gradients. Adapting this fixed number yields control over the memory requirement. This aspect is most clearly visible in Algo. 2 in A.3.6, where this number is called the “mini-batch sizes $n$”. In L.11 of Algo. 2 mini-batches are sequentially sampled (without replacement) from the buffer until the buffer is empty. Sampling until the buffer is empty is a design choice and if desired one could reduce the runtime either by reducing the buffer size (not storing the entire trajectories) or by sampling only a fraction of the total available buffer.
>
> In our work we simply stored all the diffusion steps and sampled the entire buffer (as advised in (Huang, et al., 2022). This sequential sampling (L.11 in Algo. 2) inflicts additional runtime compared to not using PPO but not using PPO would have correspondingly higher memory requirements.
>
> Experimentally, we find that for more diffusion steps using PPO is strictly superior in terms of performance, training runtime and memory requirements. This is evidenced by comparing SDDS w/ RL at 12 steps with DiffUCO at 16 steps in Fig. 1 (left) (A.8.1). While DiffUCO 16 requires 4  times more memory than SDDS w/ RL at 12 it also takes longer to train (see Tab. 7) and yields worse performance.
> We note that questions on comparisons in terms of computational resources do in general not have unambiguous answers since they depend on the precise setting of the comparison (how much time, memory, overhead for jitting, GPU type, the problem type, etc.).
>
> **Q2: “Why was PPO specifically chosen for optimizing the rKL-based objective?”**
>
> The original goal was to find a way to decouple the memory requirements in diffusion-based samplers like DiffUCO from the number of diffusion steps. RL-methods that rely on a rollout buffer yield precisely this functionality since they do not require to backpropagate through entire trajectories at once but only through mini-batches from the buffer. Among these RL-methods we selected PPO since it excels in terms of stability, sample efficiency, and in particular in terms of algorithmic simplicity compared to e.g. TRPO methods (Huang, et al., 2022).
>
> **Q3: “Can the authors provide theoretical or empirical insights into the convergence properties of the proposed methods? Specifically, under what conditions do the RL-based training procedures converge, and how stable are they across different problem instances?”**
>
> Our statement on theoretical convergence properties can be found in our response to W2. Regarding stability, we'd like to clarify an important point about our methodology: As detailed in Section 2.2, our model is designed to generalize across multiple combinatorial optimization (CO) problem instances within each dataset. This means we cannot comment on the convergence of the PPO objective for individual problem instances.
> In this context, it is worth noting, as mentioned in Section 6, that we maintained the same PPO hyperparameters across all CO problem types and datasets, with the only exception of one hyperparameter on MIS RB-large. This fact empirically showcases the stability of this method across different problems.
>
>
>
> **Q4: “How does varying the number of diffusion steps during training and inference influence the model's performance? Is there an optimal range of steps beyond which improvements plateau or diminish?”**
>
> We now added such an experiment in A.8.1, where we see that in general the compared methods improve with an increasing amount of diffusion steps. In Fig. 1 we only see a plateaued performance for SDDS: fKL w/ MC. For DiffUCO or for SDDS: rKL w/ RL the returns of increasing the number of diffusion steps tend to diminish (which is necessitated by the fact that one cannot become better than the optimal solution on a given problem instance).

---

> ### Author Response · Authors · 2024-11-22
> **Response to Review 3/3**
>
> **Q5: “Why was DiffUCO the primary(or only) comparison, and have the authors considered additional benchmarks such as those in Zhang et al. (2023)? Given that MDS results might not be state-of-the-art, how would the method perform against others besides DiffUCO?”**
> As mentioned in our response to W4 above, the comparison to DiffUCO is important since it represents an ablation of our proposed methods. By no means do we only compare to this method. In Tab. 1 and Tab. 2, we also compare to Zhang et al. (2023) and Gurobi. Due to space limitations, we only show the result of the two best ML-based methods DiffUCO (Sanokowski et al., 2024) and LTFT (Zhang et al., 2023). However, in order to emphasize that our UCO benchmarks are well established we now added extended result tables in App. 12 (Tab. 10, Tab.11 and Tab.12) which include comparisons to a large number of additional published results. Following the suggestion of reviewer RMkz we also added results and benchmarks on further spin glass models in A.8.2 and Tab. 8 and Tab. 9.
>
> Concerning additional benchmarks we would like to stress that we conduct experiments with 3 different methods on 7 different datasets UCO benchmarks and 4 different methods on two unbiased sampling benchmarks, respectively. With this rebuttal, we provide an additional comparison of three different methods in unbiased sampling (A.8.2), another experiment on the ground state prediction of spin glasses (A.8.2) and a comparison of three methods over an increasing number of diffusion steps (A.8.1). These experiments along with re-training runs for the determination of error bars exhaust our computational resources. We are not aware of any other related work that provides evaluations on such numerous and diverse benchmarks including optimization and sampling problems.
>
> Concerning the comparison to other methods on MDS we refer to the extended results table in A.12 Tab. 11. These results show that both DiffUCO and SDDS: rKL w/ RL outperform numerous other methods that were benchmarked on this problem. In particular, we would like to highlight that the next best method after SDDS and DiffUCO is LTFT, which takes orders of magnitudes longer.
>
>
>
> In conclusion, we thank the reviewer for highlighting several shortcomings in the presentation of our work and for the suggestion of a feasible and well motivated further experimental evaluation on spin glasses. We are convinced that we could use this feedback to improve our work significantly and hope that these improvements will be reflected in corresponding updates of the reviewer's rating.
>
>
>
> **References:**
>
>
>
> Hibat-Allah, M., Inack, E. M., Wiersema, R., Melko, R. G., & Carrasquilla, J. (2021). Variational neural annealing. Nature Machine Intelligence, 3(11), 952-961.
>
>
>
> Huang, et al., "The 37 Implementation Details of Proximal Policy Optimization", ICLR Blog Track, 2022.
>
>
>
> Sanokowski, S., Hochreiter, S., & Lehner, S. (2024). A Diffusion Model Framework for Unsupervised Neural Combinatorial Optimization. arXiv preprint arXiv:2406.01661.
>
>
>
> Tian, H., Olshevsky, A., & Paschalidis, Y. (2024). Convergence of actor-critic with multi-layer neural networks. Advances in neural information processing systems, 36.
>
>
>
> Wu, D., Wang, L., & Zhang, P. (2019). Solving statistical mechanics using variational autoregressive networks. Physical review letters, 122(8), 080602.
>
>
>
> Zhang, D., Dai, H., Malkin, N., Courville, A. C., Bengio, Y., & Pan, L. (2023). Let the flows tell: Solving graph combinatorial problems with gflownets. Advances in neural information processing systems, 36, 11952-11969.

---

> > ### Comment · Reviewer_RMkz · 2024-11-25
> >
> > Thank you for the detailed and thoughtful response. Most of my questions have been thoroughly addressed, and I appreciate the additional experiments and explanations provided. Based on these improvements and clarifications, I will revise my score from 5 to 6.

---

### Official Review · Reviewer_vFfF · 2024-11-04

**Soundness:** 3
**Presentation:** 3
**Contribution:** 3
**Rating:** 6
**Confidence:** 4

**Summary:**

This paper presents improvements to existing diffusion-based methods for sampling from discrete unnormalized distributions by introducing two techniques that reduce memory requirements for training the forward and reverse KL objectives. For the reverse KL objective, the authors utilize the policy gradient theorem to lower memory costs, which typically scale linearly with the number of diffusion steps in most samplers. In addressing the forward KL objective, they reformulate the loss using importance sampling and employ Monte Carlo estimation to compute the sum over diffusion steps, thereby minimizing memory usage during backpropagation. The authors validate their methods through experiments on Ising model benchmarks.

**Strengths:**

* The innovative approach to computing the forward and reverse KL loss functions is the paper's main strength. Developing memory-efficient loss functions is particularly important for diffusion models.
* The paper is clearly written and all the results are well presented.

**Weaknesses:**

* The empirical validation of the proposed algorithms is insufficient to convincingly demonstrate their practical benefits. It would be more compelling if the authors included comparisons to standard diffusion-based samplers under the same computational constraints or memory budget in the Ising model experiments. The current comparisons against the AR baselines do not adequately highlight the potential advantages of the proposed methods over conventional diffusion-based samplers.

* The experiments conducted do not appear to be sufficiently challenging, and the memory requirements of standard diffusion-based samplers in these cases may not pose significant issues in practice. It would be beneficial for the authors to test their methods in more complex settings in statistical physics (e.g., the phi-4 model) or on image generation tasks (e.g., CIFAR-10).

**Questions:**

* I am interested in the memory usage for each set of experiments reported in the paper. Including this information would enhance the presentation. Additionally, it would be valuable to see how model performance varies with the number of diffusion steps and memory budget. Specifically, I wonder if the memory budget is genuinely a bottleneck for standard diffusion-based samplers in the experiments presented.

---

> ### Author Response · Authors · 2024-11-22
> **Response to review 1/2**
>
> We thank the reviewer for the thoughtful feedback.
>
> The reviewer mentions as a weakness that our work lacks the comparison to “standard diffusion-based samplers” and “conventional diffusion-based samplers”. As we outline in the related work section there are - apart from Sanokowski et al. (2024) (DiffUCO) - to the best of our knowledge no other works on discrete diffusion-based samplers, i.e. samplers that are trained without using data from the target distribution. However, we compare DiffUCO to our approach extensively in all of our experiments under the same computational constraints (see L.384 ff., L.474 ff. L.478 ff.). Unfortunately, the reviewer does not specify which diffusion sampler other than Sanokowski et al. (2024) are requested for comparison. We note that we compare our method to numerous non-diffusion-based samplers (see the extended results tables in App. 12).
>
> In the last weakness bullet point, the reviewer suggests experiments in image generation (e.g. CIFAR-10). In this context we would like to stress that works on sampling from unnormalized distributions typically do not rely on training data in the form of samples. Hence, the learning tasks considered in this category of methods are fundamentally different from generative models which are trained using data samples. Conversely, there is no known energy function or unnormalized distribution corresponding to CIFAR-10. Consequently, these two different tasks are tackled in two separate, well established research communities. To clarify that our work is on samplers rather than generative models we point out the absence of training data in L.037 ff.
>
> This potential source of confusion on the scope of our work may also relate to the point above on the “standard diffusion-based samplers” and why we do not and cannot compare to prominent methods like e.g. DDPM (Ho et al. 2020) which are designed for generative modeling rather than sampling.
>
>
> **W: “ It would be more compelling if the authors included comparisons to standard diffusion-based samplers under the same computational constraints or memory budget in the Ising model experiments”**
>
> As mentioned above, to the best of our knowledge there is no other work on discrete diffusion sampler apart from DiffUCO. On CO benchmarks we compare our proposed methods to DiffUCO under comparable computational resources (memory and training time, see L.384 ff., L.474 ff. L.478 ff.) and show that they represent a significant improvement with respect to DiffUCO on 6 out of 7 UCO benchmarks.
>
> Concerning the unbiased sampling experiments on Ising model we state in L. 506 ff. that: “We do not report the performance of DiffUCO as this method suffered from severe mode collapse in this setting, where it only predicted either +1 or −1 for every state variable, resulting in poor behavior for unbiased sampling.”. Also for these experiments, we compare our methods to relevant other sampling methods (see Tab. 4), namely results reported in Nicoli et al. (2020) as well as our own implementation of their work, and the newly added results based on Wu et al. (2019). These results highlight that our method represents for the first time a discrete diffusion-based sampler that is applicable and highly performant in unbiased sampling.
>
> **W: “The experiments conducted do not appear to be sufficiently challenging.”**
> It is not obvious to us why these benchmarks seem to be not sufficiently challenging. We suspect that a reason may be that the result tables Tab. 1 and Tab. 2 do not show many previous works that have already targeted these benchmarks and which are no longer state-of-the-art. The choice to show only the currently strongest methods on these benchmarks is necessitated by space limitations (L.399ff). However, we added extended result tables that include these additional methods in App.12. This list of results obtained by prominent prior works shows that these benchmarks are in fact very well established precisely because they have proven over the years to remain very challenging and still allow for sizable improvements as our results show. The reviewer raises concerns on whether we evaluated on challenging and meaningful benchmarks. We took particular care to select only exactly such benchmarks.
>
> For instance, the RB graphs are designed to yield particularly hard Constraint Satisfaction Problems that are located at the satisfiability phase transition (Xu et al., 2005). They are, therefore, also a popular choice for CO benchmarks (Tönshoff et al., 2020, Wang and Li, 2023, Zhang et al., 2023, Sanokowski et al., 2024). Similarly, we selected the ferromagnetic (J=1) Ising model at the critical temperature for unbiased sampling since this seemingly simple system is known to exhibit large-range correlations that are notoriously hard to sample. Therefore, also this model is a popular choice in the corresponding literature (Wu et al. (2019), McNaughton et al., (2020), Nicoli et al. (2020), Wu et al. (2021)).

---

> ### Author Response · Authors · 2024-11-22
> **Response to review 2/2**
>
> **W: “ It would be beneficial for the authors to test their methods in more complex settings in statistical physics (e.g., the phi-4 model) or on image generation tasks (e.g., CIFAR-10).”**
> Our work contributes to the established literature on discrete sampling with concrete examples in UCO and unbiased sampling on a discrete system and as suggested by reviewer **UBbm** we have now added an additional experiment in unbiased sampling on spin glasses in A.8.2.
>
> The phi-4 model is a paradigmatic example for a field theory and is, therefore, intrinsically continuous. Hence, we it is not a suitable benchmark for this work. Adapting our work to continuous domains is an ongoing project which gives rise to new challenges, like the numerical stability of the entropy regularization for continuous domains.
>
> Similarly, the proposed experiments on CIFAR-10 are not within the scope of our work since we propose a sampling method, i.e. we work in the data-free setting (as stated in L.37ff). We outlined above the distinction between sampling methods and generative models.
>
> **Q: “I am interested in the memory usage for each set of experiments reported in the paper. Including this information would enhance the presentation.”**
>
> We added details on the utilized hardware including the available memory in A.9.
>
> **Q: “Additionally, it would be valuable to see how model performance varies with the number of diffusion steps and memory budget. Specifically, I wonder if the memory budget is genuinely a bottleneck for standard diffusion-based samplers in the experiments presented.”**
>
> In Figure 1 (A.8.1), we compare three approaches: DiffUCO, SDDS: fKL w/ MC, and SDDS rKL w/ RL. We evaluate these methods across an increasing number of diffusion steps, with marker sizes indicating memory requirements. Our findings reveal that without memory constraints, DiffUCO's performance improves as we increase the number of diffusion steps. However, under memory constraints, we cannot rely on DiffUCO to improve model performance through increased diffusion steps. Instead, SDDS rKL w/ RL proves to be the better choice, as its memory requirements remain constant regardless of the number of diffusion steps. This advantage is reflected in Figure 1, where SDDS rKL w/ RL outperforms DiffUCO in 6 out of 8 cases.
>
> Given these clarifications about our work and the extended experimental evaluations, we respectfully ask the reviewer to consider updating their score to better reflect our work's contributions.
>
>
>
>
> **References:**
>
> Sanokowski, S., Hochreiter, S., & Lehner, S. (2024). A Diffusion Model Framework for Unsupervised Neural Combinatorial Optimization. arXiv preprint arXiv:2406.01661.
>
> Ho, Jonathan, Ajay Jain, and Pieter Abbeel. "Denoising diffusion probabilistic models." Advances in neural information processing systems 33 (2020): 6840-6851.
>
> Xu, Ke, et al. "A simple model to generate hard satisfiable instances." arXiv preprint cs/0509032 (2005).
>
> Toenshoff, Jan, et al. "Graph neural networks for maximum constraint satisfaction." Frontiers in artificial intelligence 3 (2021): 580607.
>
> Wang, Haoyu Peter, and Pan Li. "Unsupervised Learning for Combinatorial Optimization Needs Meta Learning." The Eleventh International Conference on Learning Representations.
>
> Wu, Dian, Lei Wang, and Pan Zhang. "Solving statistical mechanics using variational autoregressive networks." Physical review letters 122.8 (2019): 080602.
>
> McNaughton, B., et al. "Boosting Monte Carlo simulations of spin glasses using autoregressive neural networks." Physical Review E 101.5 (2020): 053312.
>
> Nicoli, Kim A., et al. "Asymptotically unbiased estimation of physical observables with neural samplers." Physical Review E 101.2 (2020): 023304.
>
> Wu, Dian, Riccardo Rossi, and Giuseppe Carleo. "Unbiased Monte Carlo cluster updates with autoregressive neural networks." Physical Review Research 3.4 (2021): L042024.

---

> > ### Comment · Reviewer_vFfF · 2024-11-25
> >
> > I thank the authors for the efforts they have made on the revision and I am satisfied with their replies. So I am raising my score to 6.

---

### Author Response · Authors · 2024-11-22
**General Response:**

We thank all reviewers for their thoughtful and constructive critique. We are glad to see that the contribution and soundness of our work are well received. Nevertheless, following the reviewers’ feedback we made numerous significant improvements which include but are not limited to:

-   modification of several parts of the main text (highlighted in red font color) for improved understandability and readability,

-   adding a comprehensive study of efficiency in terms of training time, memory, and number of diffusion steps in order to highlight the practical significance of our work in terms of its superior scalability which demonstrably underlies its superior performance (A.8.1),

-   extension of our experimental evaluation with challenging spin glasses on which we demonstrate unbiased sampling and optimization capabilities and compare to corresponding prior work (A.8.2),

-   a presentation of extended result tables in A.12 to emphasize that our benchmarks are very well established in the current literature.

---

### Meta-Review · Area_Chair_mxhf · 2024-12-19

**Metareview:**

The paper proposes better objectives for learning diffusion processes that sample from a given discrete distribution (defined as a probability function). Namely, the issue that this paper aims to solve is the need to backpropagate through simulated trajectories, which the authors propose to address using either Importance Sampling (forward KL objective) or Proximal Policy Optimization (reverse KL objective). The authors demonstrate the utility of their approach sampling from the Ising model.

All the reviewers acknowledge the importance of the problem and the contribution of the paper. Several concerns have been raised regarding the experimental setup considered, but the authors managed to clarify these concerns in the rebuttal. Despite the absence of strong acceptance, I believe that the problem considered and the proposed solution would be of interest to the ICLR community.

**Additional Comments On Reviewer Discussion:**

The main concern raised by all the reviewers was the choice of task. Namely, the authors demonstrate the utility of their method for the Ising model, which is not a common problem at ICLR (hence, the concerns of the reviewers). Despite that, the authors managed to sway the initial opinion of the reviewers about the problem by providing additional experiments (on another physics model) and providing extensive literature references.

---

### Decision · Program_Chairs · 2025-01-22

Accept (Poster)